# Chemotaxis-driven delivery of nano-pathogenoids for complete eradication of tumors post-phototherapy

Min Li[1,2,3,5], Shuya Li[1,5], Han Zhou[1], Xinfeng Tang[1], Yi Wu[1], Wei Jiang[1], Zhigang Tian [1], Xuechang Zhou [3], Xianzhu Yang[4] & Yucai Wang[1,2 ✉]

The efficacy of nano-mediated drug delivery has been impeded by multiple biological barriers such as the mononuclear phagocyte system (MPS), as well as vascular and interstitial barriers. To overcome the abovementioned obstacles, we report a nano-pathogenoid (NPN) system that can in situ hitchhike circulating neutrophils and supplement photothermal therapy (PTT). Cloaked with bacteria-secreted outer membrane vesicles inheriting pathogen-associated molecular patterns of native bacteria, NPNs are effectively recognized and internalized by neutrophils. The neutrophils migrate towards inflamed tumors, extravasate across the blood vessels, and penetrate through the tumors. Then NPNs are rapidly released from neutrophils in response to inflammatory stimuli and subsequently taken up by tumor cells to exert anticancer effects. Strikingly, due to the excellent targeting efficacy, cisplatin-loaded NPNs combined with PTT completely eradicate tumors in all treated mice. Such a nano-platform represents an efficient and generalizable strategy towards in situ cell hitchhiking as well as enhanced tumor targeted delivery.

[1] Division of Molecular Medicine, Hefei National Laboratory for Physical Sciences at Microscale, The CAS Key Laboratory of Innate Immunity and Chronic Disease, School of Life Sciences, University of Science and Technology of China, 230027 Hefei, Anhui, China. [2] Intelligent Nanomedicine Institute, The First Affiliated Hospital of USTC, Division of Life Sciences and Medicine, University of Science and Technology of China, 230001 Hefei, Anhui, China. [3] College of Chemistry and Environmental Engineering, Shenzhen University, 518060 Shenzhen, Guangdong, China. [4] Institutes for Life Sciences, School of Medicine and National Engineering Research Center for Tissue Restoration and Reconstruction, South China University of Technology, 510006 Guangzhou, China. [5] These authors contributed equally: Min Li, Shuya Li. ✉email: yucaiwang@ustc.edu.cn

The initiation and progression of cancer are often associated with chronic inflammation and certain cancer treatment modalities have been demonstrated to induce acute inflammation[1–4]. Once inflammation is elicited, neutrophils are the key cell type that is capable of actively homing to the sites of tissue damage or infection, extravasating across the endothelial lining, and penetrating deep into nonvascular regions, following the concentration gradient of chemokines, a process known as chemotaxis[5].

On the other hand, synthetic nanoparticles (NPs) do not have the chemotactic feature and also cannot easily pass common biological barriers because of its artificial nature[6]. For example, NPs can be rapidly opsonized and thoroughly cleared by the mononuclear phagocyte system (MPS) such as those in the liver and spleen[7,8]. Next, the NP population evading MPS clearance needs to extravasate out of the blood circulation and cross the endothelial cells that outline the tumor vasculature[9]. Furthermore, the tumor interstitium featuring high-interstitial fluid pressure and dense extracellular matrices would also prevent the diffusion and penetration of NPs into the tumoral microspace[10,11].

To overcome these hurdles for synthetic NP delivery, neutrophils have been utilized as "self by host" natural carriers of drugs or NPs for nano-mediated therapy[12–15]. The premise of utilizing neutrophils to transport NPs is the construction of the NP-neutrophil complex, which is largely based on two distinct strategies. The first one is to assemble NPs into neutrophils ex vivo, and the second one is to hitchhike neutrophils in situ (i.e., during blood circulation) using NPs[4,16,17]. In practice, the former suffers from reduced cell viability due to the short lifetime of neutrophils (~7 h), premature intracellular degradation of the cargos, high cost, insufficient quantities of harvested cells, as well as the risk of in vitro contamination. The in situ hitchhiking strategy is more advantageous towards clinical practice, and the rational design of NPs with high-binding affinity to neutrophils is required. To this end, Wang and colleagues[16,18,19] reported the use of denatured bovine serum albumin NPs and anti-CD11b antibody-decorated NPs to target circulating neutrophils for the treatment of lung infections and cancers. Nevertheless, the development of more generalizable and efficient methods to specifically hitchhike circulating neutrophils remains largely unexplored and challenging.

Herein, we propose a strategy for the use of pathogen-mimicking nano-pathogenoids (NPNs) to hitchhike circulating neutrophils. The pathogen-mimicking concept was derived from the manner in which neutrophils in nature fight invading pathogens, such as bacteria, fungi, and viruses. Neutrophils are known to sense and ingest pathogens through the recognition of pathogen-associated molecular patterns (PAMPs)[5,20]. By cloaking NPs with bacteria-secreted outer membrane vesicles (OMVs), one would be able to create a nano-sized mimic of the parent bacteria with similar pathological activities[21]. Meanwhile, since the synthetic NPNs are non-replicating, potential toxicity issues could also be less when compared to live or weakened bacteria[22–25].

As a proof-of-principle demonstration, we explored the utility of NPNs for the elimination of residual microtumors after photothermal therapy (PTT). We chose to supplement PTT because the heat generation in the tumor tissue could naturally create an acute inflammatory environment for the recruitment of neutrophils with our NPNs from the blood circulation[26]. In addition, PTT is known to suffer from tumor recurrence due to the incomplete eradication of microtumors by the shallow light penetration depth in the tumor tissue[27]. It is therefore of great importance to develop combinatorial therapeutic methods to supplement the PTT efficacy. In this work, we demonstrate that the local heating in the residual microtumors can actually broadcast their individual locations and subsequently trigger the migration of the neutrophils (Fig. 1). Importantly, intravenously (i.v.) administered NPNs are recognized and internalized by neutrophils and delivered to the residual microtumors along with homing of neutrophils. Using this strategy, the tumor accumulation of NPNs is doubled when compared to NPs delivered solely by the passive targeting, i.e., the enhanced permeability and retention (EPR) effect. After reaching the tumor tissue, NPNs are released from neutrophils upon the formation of neutrophil extracellular traps (NETs) and then be ingested by residual tumor cells for intracellular functions. Strikingly, the treatment of PTT plus cisplatin-loaded NPNs completely eradicate tumors of all the treated mice. Overall, these findings hold promise for the use of pathogen-mimicking NPNs for the in situ hitchhiking of circulating neutrophils and subsequent enhanced tumor targeting.

## Results

**Chemotaxis of neutrophils to PTT primed tumors.** Initially, we investigated whether localized PTT treatment could cause an inflammatory tumor microenvironment and subsequent neutrophil infiltration. A donor-acceptor conjugated polymer PBIBDF-BT (with alternating isoindigo derivative [BIBDF] and bithiophene [BT]) with optical absorption that peaked at 820 nm was used as the PTT transducer[28]. To enable the systematic administration and efficient tumor accumulation of PBIBDF-BT, it was encapsulated into the core of a long-circulating micelle composed of an FDA-approved polymer poly(ethylene glycol)-*b*-poly(lactic-co-glycolic acid) (PEG-*b*-PLGA) to produce NPs@PBT (Supplementary Fig. 1). The NPs@PBT had an average diameter of $168.3 \pm 3.0$ nm (Fig. 2a) and zeta potential of $-2.5 \pm 1.7$ mV (Fig. 2b). To better control the temperature rise upon laser irradiation, an 808 nm laser was connected to a signal generator to regulate its output frequency (Fig. 2c). As a proof-of-principle, the PTT temperatures of the NPs@PBT aqueous suspensions could be constantly controlled at 40, 42, and 45 °C by carefully regulating the duty cycle for a square waveform of the signal generator, which was set as 65%, 75%, and 90% ON, respectively (Fig. 2d).

To study the effects of PTT on neutrophil infiltration, EMT6 (murine breast carcinoma cells) tumor-bearing mice were *i.v.* injected with NPs@PBT and the temperatures of the PTT-treated tumors were maintained at 40 °C for 5 min (Fig. 2e). The time course of neutrophil percentages (relative to total CD45$^+$ leukocytes) in the tumor and blood were analyzed using flow cytometry; typical gating strategies were shown in Supplementary Fig. 2. A significant increase in CD45$^+$CD11b$^+$Ly6G$^+$ neutrophils in the tumor was observed within 48 h post-PTT and peaked at 4 h (Fig. 2f, g and Supplementary Fig. 3). In line with this, a similar increase in neutrophils in the blood was also observed within the first 12 h followed by a gradual decrease over a period of 96 h (Fig. 2h and Supplementary Fig. 4), which supported the rapid recruitment of tumor infiltrating neutrophils from blood induced by mild PTT.

We next studied the effect of tumor size on neutrophil infiltration. PTT (40 °C for 5 min) was applied to mice bearing EMT6 tumors of different volumes (tumor volume ~60, 100, and 240 mm$^3$). An abundance of infiltrating neutrophils in PTT-treated tumors was found in all three groups, but the recruitment efficiency decreased with the increase of tumor size (Fig. 2i). The phenomenon could be due to insufficient tissue penetration of near-infrared (NIR) light into the larger tumors, which was evidenced by the less apoptosis and neutrophil recruitment as well as higher cell proliferation rate in the deeper regions of the 240 mm$^3$ tumors compared to the 60 mm$^3$ ones (Supplementary Fig. 5). In addition, blood vessel congestion and occlusion in the

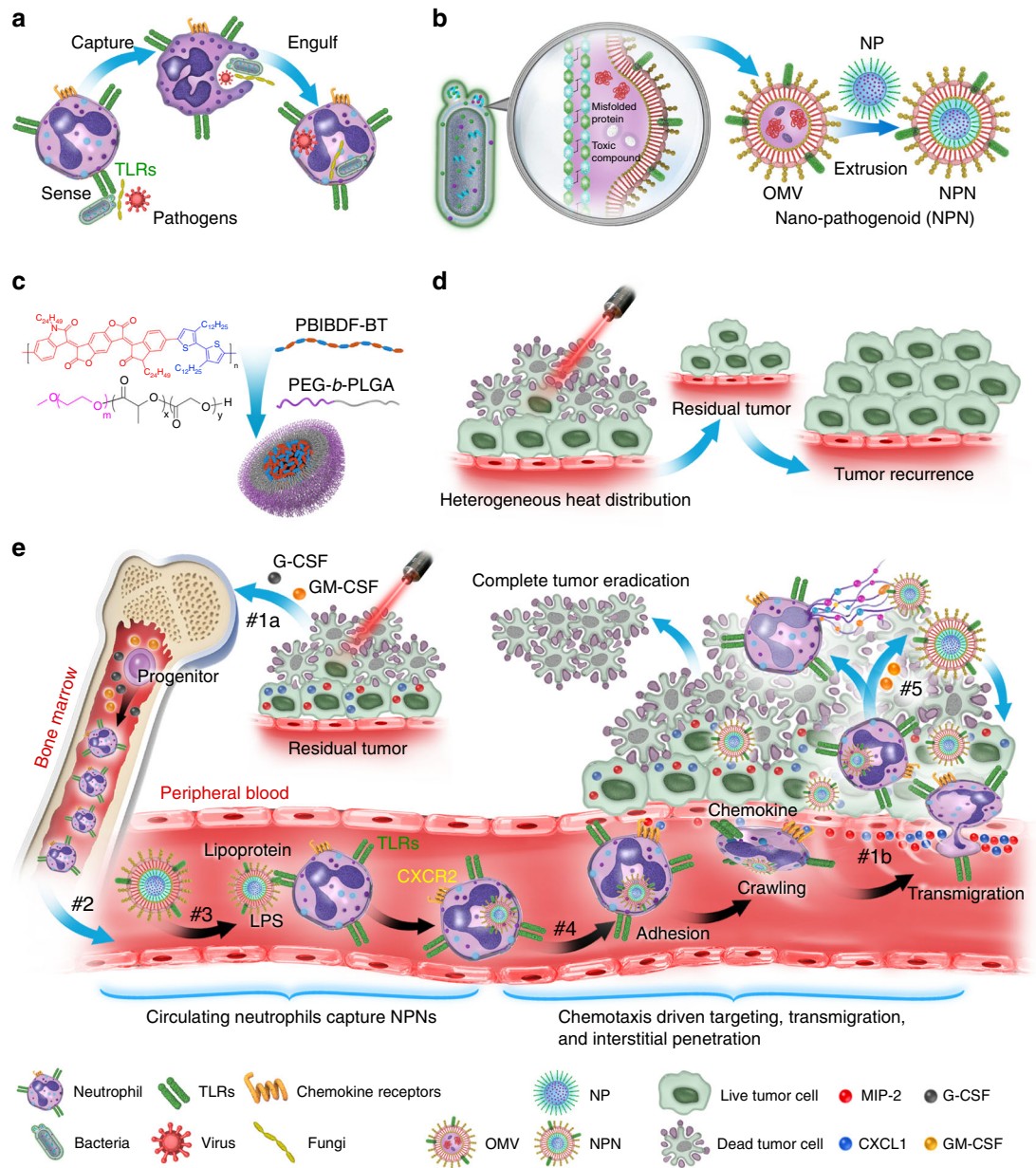

**Fig. 1 Schematic illustration showing the chemotaxis-driven delivery of NPNs for complete eradication of tumors post-phototherapy. a** Neutrophils sense, capture, and engulf pathogens by recognizing the PAMPs with toll-like receptors (TLRs). **b** Preparation of NPNs by coating OMVs on NPs, which inherit PAMPs from the OMVs. **c** Preparation of PEG-*b*-PLGA NPs encapsulating PBIBDF-BT (PBT) as a photothermal transducer. **d** The limited penetration of laser light used in PTT causes heterogeneous heat distribution within the tumor tissue and incomplete eradication of tumors, thus leading to tumor recurrence. **e** Treatment-induced cell death created an inflammatory environment of the residual tumor and induced the production of granulocyte colony stimulating factor (G-CSF), granulocyte-macrophage colony stimulating factor (GM-CSF), and chemokines CXCL1 and MIP-2. #1a The released G-CSF and GM-CSF increased neutrophil production from bone marrow. #1b The released CXCL1 and MIP-2 broadcasted the location of the inflamed tumor. #2 Neutrophils entered the blood circulation and encountered the injected NPNs. #3 Neutrophils sensed NPNs with the recognition of LPS and lipoprotein by TLRs and subsequently engulfed them. #4 Neutrophils laden with NPNs were recruited into the tumor site in response to the chemokine gradient through the following cascade: adhesion, crawling and transmigration. #5 NPNs were released from neutrophils to kill tumor cells along with the formation of NETs in the inflamed tumor.

core of larger tumors might also hinder neutrophil recruitment[29]. We further extended the studies to a CT26 syngeneic colon carcinoma model and observed similar neutrophil infiltration, suggesting that the PTT-induced recruitment of neutrophils was not tumor type-dependent (Fig. 2i and Supplementary Fig. 6). In addition, higher PTT temperature (Fig. 2j) and extended irradiation time (Fig. 2k) could cause more neutrophil infiltration, probably because of the more severe local inflammation. Accordingly, PTT of 40 °C was mainly used in the rest of other

in vivo experiments since such a mild temperature was sufficient to create an acute intratumoral inflammation for neutrophil recruitment and PTT of higher temperatures has been reported to cause severe side effects in normal tissues near the lesions[30]. Furthermore, PTT using metallic gold nanorods (GNRs) as the transducer led to similar tumor accumulation of neutrophils, suggesting that neutrophil infiltration induced by PTT was independent on the type of transducers (Supplementary Fig. 7). The chemokine C-X-C motif ligand 1 (CXCL1) and macrophage-inflammatory protein-2

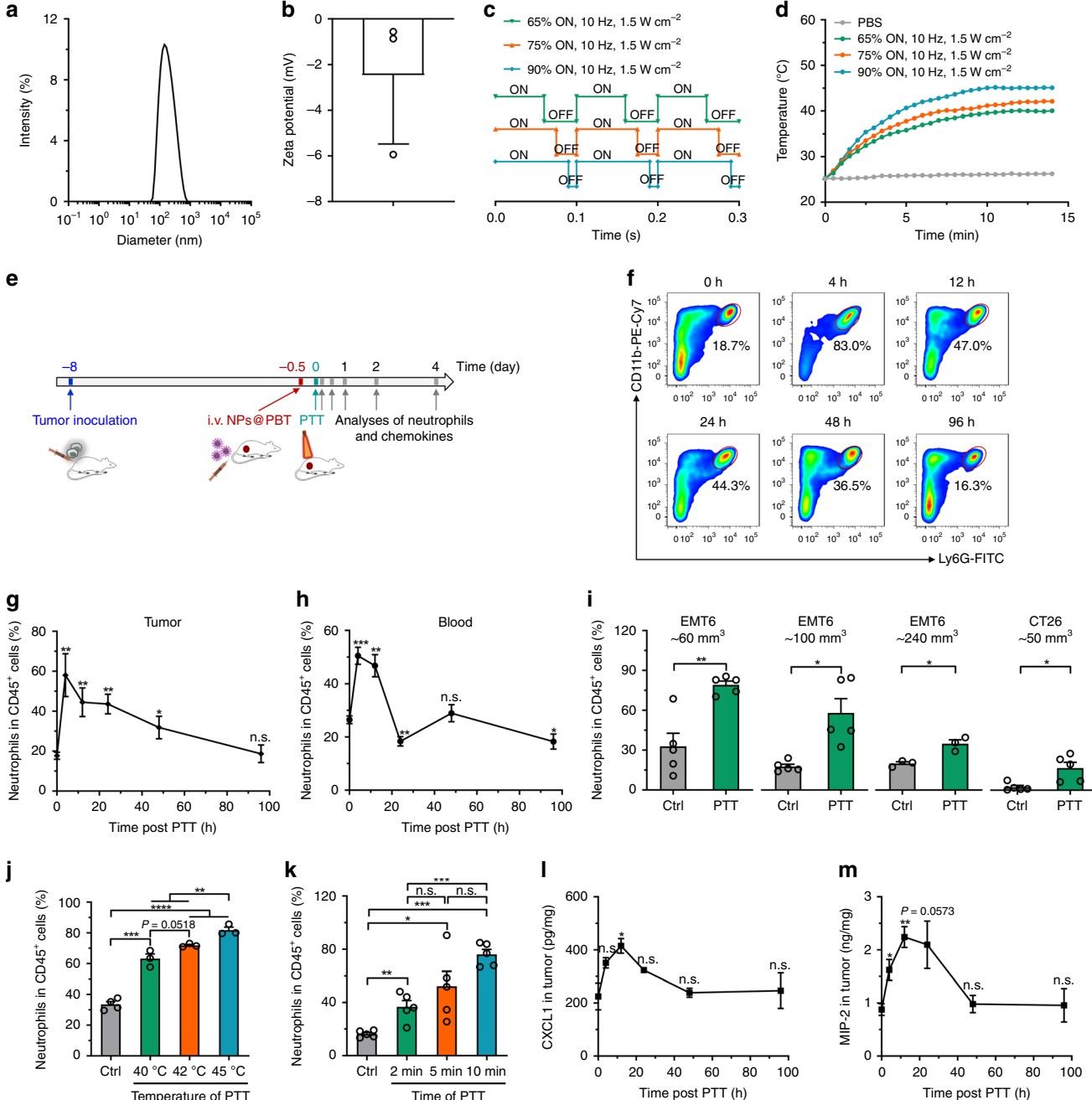

**Fig. 2 PTT caused inflammatory microenvironment in tumors and induced infiltration of neutrophils. a** Size and **b** zeta potential of NPs@PBT. $n = 3$ biologically independent samples. **c** Duty cycles for square wave input of a waveform generator connected to an 808 nm laser. **d** Temperature change curves of NPs@PBT suspensions under laser irradiation. **e** Time schedule for measuring the dynamic change of neutrophils and chemokines in tumor and blood post-PTT. Mice bearing EMT6 or CT26 tumors were *i.v.* injected with NPs@PBT (−12 h) and the tumors were subjected to PTT at 40, 42 and 45 °C for 2, 5, and 10 min, respectively. At the indicated time points, the tumors and blood were harvested for analysis. **f** Flow cytometric measurements and **g** corresponding quantitative analysis of neutrophil (CD11b$^+$Ly6G$^+$) percentage in DAPI$^-$CD45$^+$ tumor-infiltrating leukocytes at the indicated time points post PTT (40 °C for 5 min). $n = 5$ per time point. **h** Neutrophil (CD11b$^+$Ly6G$^+$) percentage in CD45$^+$ blood leukocytes at different time points post-PTT (40 °C for 5 min). $n = 5$ per time point. **i** Neutrophil (CD11b$^+$Ly6G$^+$) percentage in DAPI$^-$CD45$^+$ tumor-infiltrating leukocytes in different types of tumors with varied tumor sizes at 4 h post-PTT (40 °C for 5 min). $n = 5$ per group for EMT6 ~60, 100 mm$^3$, and CT26 ~50 mm$^3$, $n = 3$ per group for EMT6 ~240 mm$^3$. **j, k** Neutrophil (CD11b$^+$Ly6G$^+$) percentage in DAPI$^-$CD45$^+$ tumor-infiltrating leukocytes in EMT6 tumors 4 h post-PTT with varied **j** temperatures and **k** laser irradiation time. For **j**, $n = 4$ for control group, and $n = 3$ for all other groups. For **k**, $n = 5$ per group. **l, m** ELISA analyses of the contents of **l** CXCL1 and **m** MIP-2 in EMT6 tumors at the indicated time points post-PTT (40 °C for 5 min). $n = 3$ per group. Data are shown as mean ± SEM and analyzed by unpaired two-tailed Student's *t*-test. \*$P < 0.05$, \*\*$P < 0.01$, \*\*\*$P < 0.001$, \*\*\*\*$P < 0.0001$. *n.s.*, not significant. Source data are provided as a Source Data file.

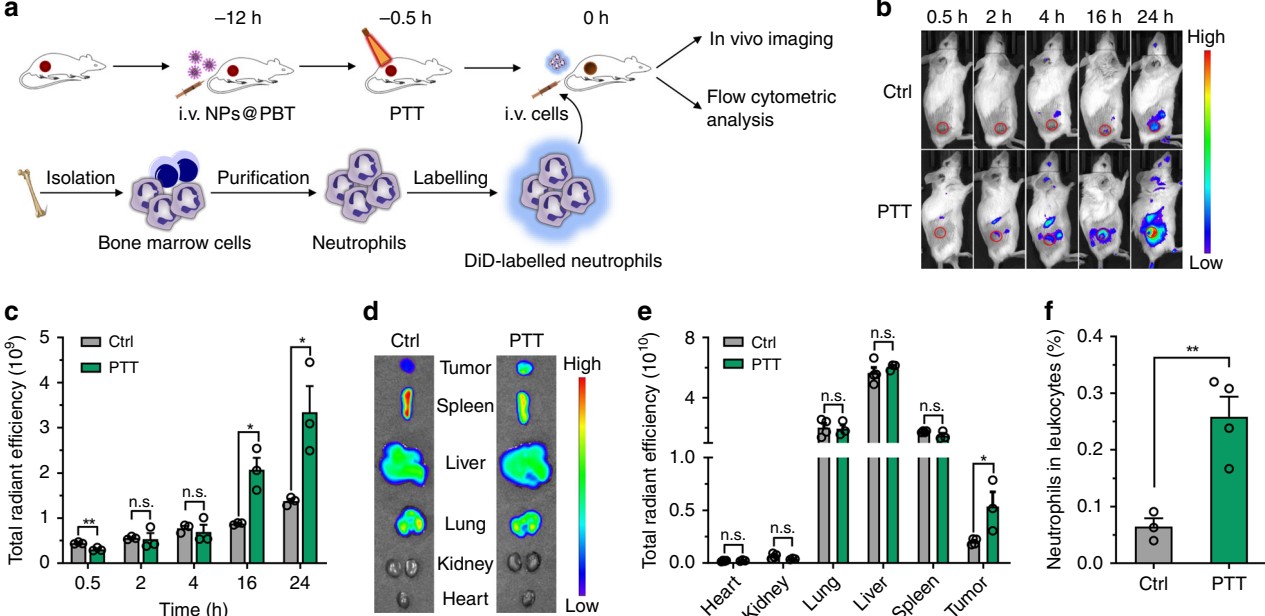

**Fig. 3 PTT-created tumoral inflammatory microenvironment recruited adoptively transferred neutrophils. a** Schematic showing analysis of the accumulation of transferred DiD-labeled neutrophils in PTT-treated EMT6 tumors. EMT6-bearing mice were *i.v.* injected with NPs@PBT at −12 h and tumors were irradiated with an 808 nm laser (40 °C for 5 min) at −0.5 h. Neutrophils were isolated from bone marrow, labeled with DiD, and *i.v.* injected into the mice on 0 h. **b, c** The distribution of transferred DiD+ neutrophils in the recipient mice was analyzed with the IVIS at the indicated time points. **b** Representative images and **c** corresponding quantification of tumoral fluorescence intensities at the indicated time points. $n = 3$ mice per group. **d** Ex vivo fluorescence images and **e** corresponding quantification of total fluorescence intensities of tumors and major organs collected at 24 h post-cell transfer. $n = 4$ for control group and $n = 3$ for PTT group. **f** The percentage of transferred DiD+ neutrophils in total DAPI−CD45+ tumor-infiltrating leukocytes analyzed by flow cytometry at 10 h post injection. $n = 3$ for control group and $n = 4$ for PTT group. Data are shown as mean ± SEM and analyzed by unpaired two-tailed Student's *t*-test. *$P < 0.05$, **$P < 0.01$. n. s., not significant. Source data are provided as a Source Data file.

(MIP-2) are two major chemoattractants responsible for recruiting neutrophils by binding to the chemokine receptor CXCR2[31]. Both chemokines were rapidly elevated in the tumors in response to PTT (Fig. 2l, m). Moreover, blocking the interaction of CXCR2 with CXCL1 and MIP-2 using a small molecule CXCR2 inhibitor SB225002 significantly reduced the neutrophil recruitment (Supplementary Fig. 8)[5,31–33]. Collectively, neutrophils migrated to PTT-treated tumors along chemokine gradients of CXCL1 and MIP-2, irrespective of the type of tumors and PTT transducers, as well as the tumor size.

**PTT-treated tumors recruited adoptively transferred neutrophils.** We further investigated the neutrophil recruitment to PTT-treated tumors using adoptively transferred neutrophils. Neutrophils were isolated from bone marrow and purified by density-gradient centrifugation with a purity of ~92% (Supplementary Fig. 9) and a viability of ~96% (Supplementary Fig. 10). The purified neutrophils preserved their basic characteristics, including a high-ingesting activity of particles (Supplementary Fig. 11) and enhanced expression of CD44 and CD11b in response to inflammatory molecules such as lipopolysaccharide (LPS) (Supplementary Fig. 12). The cells were labeled with a lipophilic fluorescent dye 1,1′-dioctadecyl-3,3,3′,3′-tetra-methylindodicarbocyanine perchlorate (DiD) with NIR emission (Em = 663 nm) and *i.v.* injected into EMT6 tumor-bearing mice at 0.5 h after PTT (Fig. 3a). The transferred neutrophils exhibited superior tumor targeting in PTT-treated EMT6-bearing mice than in untreated mice observed using an in vivo imaging system (IVIS) (Fig. 3b and Supplementary Fig. 13). Further quantitative region of interest (ROI) analysis revealed a ~2.4-fold higher DiD fluorescence signal in the PTT-treated tumors than in the control tumors at both 16 and 24 h (Fig. 3c). In addition, direct ex vivo

imaging of the excised tumors and major organs showed that PTT resulted in a ~2.7-fold increase in neutrophil recruitment to tumors without causing significant differences in other organs (Fig. 3d, e). We also analyzed the percentages of transferred cells in tumor-infiltrating leukocytes by flow cytometry 10 h post-adoptive transfer. The results indicated that the PTT-induced inflammatory microenvironment recruited ~4.0-fold more neutrophils (i.e., DiD positive) to the tumors, as compared to the untreated controls (Fig. 3f). These collective results confirmed that PTT-induced inflammation was able to recruit neutrophils to tumors in a treatment-specific manner.

**Chemotaxis-mediated neutrophil trafficking overcame biological barriers in drug delivery.** We then investigated whether the chemotaxis of neutrophils to tumors could overcome certain biological drug delivery barriers through specific targeting to disease sites and active transport in interstitial spaces. We applied intravital microscopy of a dorsal-skin-fold window chamber tumor model to real-time observe the migration behaviors of neutrophils in PTT-treated tumors. After PTT at 40 °C for 5 min, the mice were *i.v.* injected with PE-labeled anti-mouse Ly6G antibody and 3,3′-dioctadecyloxacarbocyanine perchlorate (DiO)-labeled PEG-*b*-PLGA micelles (NPs@DiO) to visualize neutrophils and blood vessels, respectively (Fig. 4a). Within 2 h after PTT, abundant neutrophils (120 neutrophils per mm² of vessels) were recruited to tumors (Fig. 4b and Supplementary Movie 1), whereas few neutrophils (21 neutrophils per mm² of vessels) appeared in the tumor vessels of the untreated mice. Specifically, in enlarged region I, neutrophils moved slower in blood vessels, and occasionally adhered to the vessel wall, with their morphology changing from an oval to an irregular shape (Fig. 4c). In enlarged region II, neutrophils were deformed into an

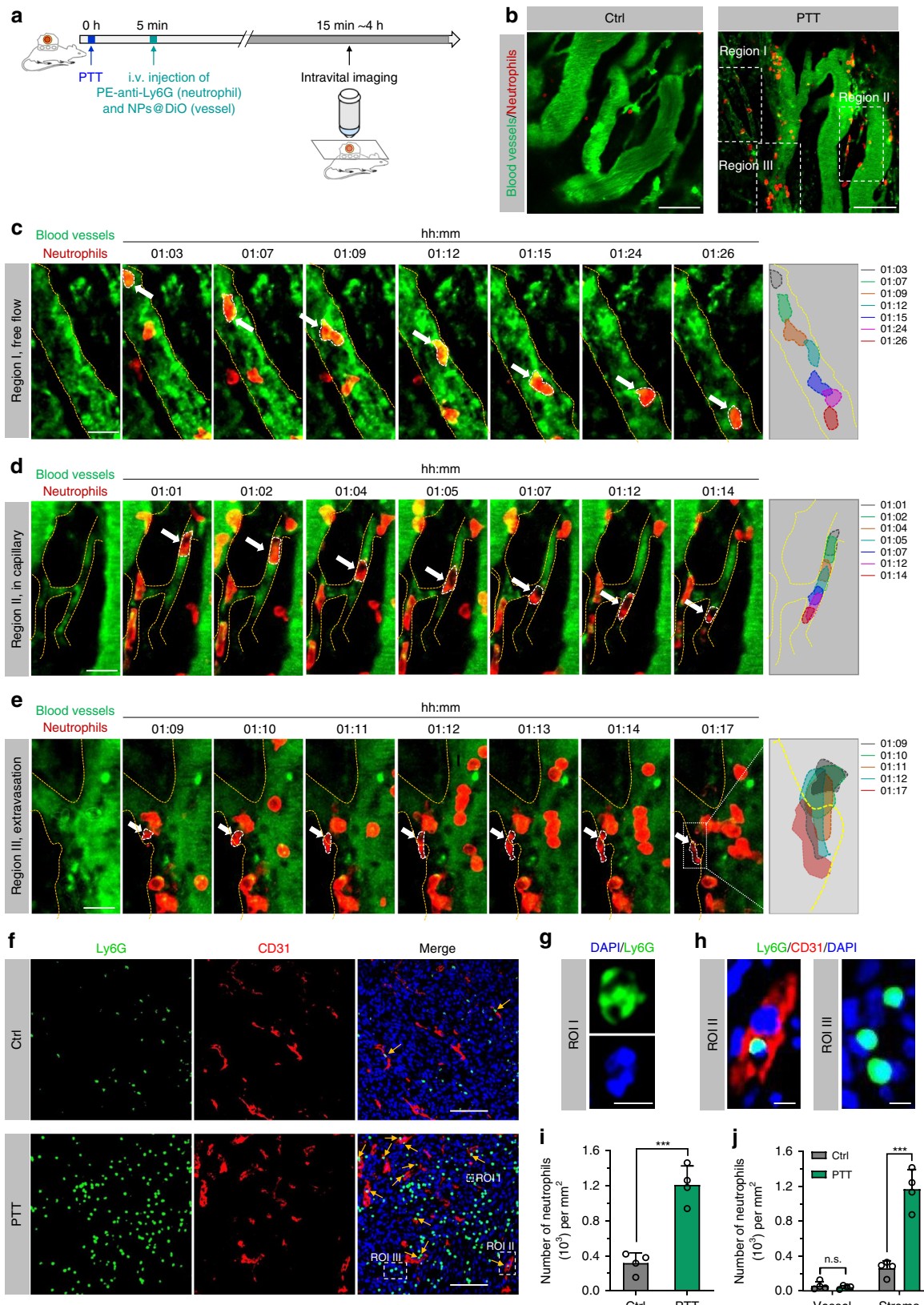

elongated shape to facilitate their movement in the small capillary, which would enable a homogenous distribution in the whole tissue (Fig. 4d). Furthermore, the neutrophil on the blood vessel elongated and squeezed through endothelial cells into the interstitium, which was completed within several minutes (Fig. 4e). Thereby, the chemokine gradient in PTT-treated tumors from the interstitial stroma to the blood would enable neutrophils to overcome the biological barriers encountered in conventional drug delivery.

The tumors were excised at 4 h after PTT for immunofluorescence (IF) imaging (Fig. 4f). We used CD31 as a blood vessel marker to assess the distribution of neutrophils (Ly6G$^+$ and

**Fig. 4 Intravascular migration, transmigration, and interstitial migration of neutrophils in PTT-treated tumors. a** Schematic showing intravital microscopy imaging of a dorsal-skin-fold window chamber EMT6 tumor model that received PTT. After 5 min, the mice were *i.v.* injected with PE-Ly6G antibody and DiO-labeled PEG-*b*-PLGA NPs (NPs@DiO) to visualize neutrophils and blood vessels, respectively. **b** Representative intravital microscopy images showed neutrophils in tumors with and without PTT treatment. Scale bars, 100 μm. **c–e** Time-lapse images showing **c** movement in blood vessels, **d** intra-capillary crawling, and **e** transmigration of neutrophils in the tumors post-PTT. The images of **c–e** were adopted from region I, II, and III in **b**, respectively. Right panels of **c–e** showing the movement track of neutrophils at indicated time points. Scale bars, 20 μm. **f** Representative IF images showed neutrophils (Ly6G) in blood vessels (CD31) and the interstitial stroma of tumors collected 4 h after PTT. Yellow arrows indicated neutrophils in blood vessels. Scale bars, 100 μm. **g** Enlarged ROI I of panel **f** presented a typical segmented nuclear morphology of neutrophils (Ly6G⁺). Scale bar, 5 μm. **h** Enlarged ROI II and ROI III from panel **f** showed neutrophils in blood vessel and tumor stroma, respectively. Scale bars, 5 μm. **b–h** Data are representative of three biological replicates. **i** The density of neutrophils in tumors before and after PTT treatment. $n = 4$ per group. **j** The density of neutrophils in blood vessels and tumor stroma. $n = 4$ per group. Data are shown as mean ± SD and analyzed by unpaired two-tailed Student's *t*-test. ***$P < 0.001$. *n.s.*, not significant. Source data are provided as a Source Data file.

exhibited segmented cell nuclei as shown in ROI I in Fig. 4g) in the vessels and in the tumor interstitium, respectively (Fig. 4h). The number of neutrophils accumulated in the PTT-treated tumors was ~3.8 times greater than that in the control tumors without treatment, due to the tumor-specific chemotaxis (Fig. 4i). Closer analysis of the intratumoral distribution of the neutrophils showed that more neutrophils were in the interstitial stroma of the PTT-treated tumors (Fig. 4j), further demonstrating the extravasation of neutrophils from the blood vessels. Taken together, these results demonstrated that recruited neutrophils were capable of crossing the blood vessel barrier and distributing in the interstitium of tumor tissues.

**Preparation and characterization of NPNs.** Encouraged by the capability of localized PTT treatment on tumors to recruit neutrophils from blood, we next sought to explore whether pathogen-mimicking NPNs could specifically hitchhike the circulating neutrophils. We collected OMVs from *Escherichia coli* (*E. coli*) and coated the OMVs onto PEG-*b*-PLGA micellar NPs to prepare NPNs by repeated extrusion through a 200 nm pore membrane (Fig. 5a). We determined that 1.0 mg of PEG-*b*-PLGA NPs could be completely coated by 0.2 mg OMVs based on a previously reported calculation method[34]. This was verified by the rise of zeta potential from $-18.13 \pm 0.47$ mV for bare NPs to $-4.16 \pm 0.32$ mV for the obtained NPNs (i.e., OMV: NPs = 1: 5, w/w), which was comparable to that of pristine OMVs ($-3.96 \pm 1.18$ mV) (Fig. 5b and Supplementary Fig. 14). The diameter of NPs increased from $142.0 \pm 3.5$ to $149.0 \pm 3.1$ nm upon coating with OMVs (i.e., NPNs) (Fig. 5c), and this size increase (about 7 nm) was consistent with the reported thickness of the lipid layer[24,34–36]. Successful coating of OMVs on NPs was verified by transmission electron microscopy (TEM) observation, showing that NPNs had a typical core-shell structure (Supplementary Fig. 15). The NPNs were stable in the biological environment as evidenced by the constant size upon serum incubation (Fig. 5d).

**NPNs hitchhiked circulating neutrophils.** To study whether NPNs could hitchhike circulating neutrophils, we *i.v.* injected DiO-labeled NPs and NPNs into PTT-treated EMT6-bearing mice. At 4 h, about 41% of the neutrophils in the blood contained NPNs, while only 10% of the neutrophils internalized the NPs (Fig. 5e and Supplementary Fig. 16). In addition, the geometric mean fluorescence intensity (GMFI) of DiO for the NPNs in the neutrophils was significantly higher than that for the NPs (Fig. 5f). Besides neutrophils, we further analyzed the distribution of NPNs in different subsets of immune cells in blood by multiplying the percentage of immune cells with their cellular GMFI of DiO. The results showed that more NPNs distributed in neutrophils and close analysis of major immune cell types in blood revealed that neutrophils internalized 83.4% of the total

NPNs, compared to 9.6, 2.6, 3.3, and 1.1% for monocytes, T cells, B cells, and NK cells, respectively (Fig. 5g).

Toll-like receptors TLR2 and TLR4 are known to mediate response of neutrophils to bacteria by recognizing the lipoprotein and LPS on bacteria, respectively[37]. In TLR4 knockout mice ($Tlr4^{-/-}$), the uptake of NPNs by circulating neutrophils was significantly inhibited (Supplementary Fig. 17a, b). In addition, blockade of TLR2 using anti-TLR2 antibody decreased the neutrophil targeting efficiency by 25% (Supplementary Fig. 17c, d). The results suggested that both receptors, especially TLR4, played a predominant role in the recognition of NPNs by neutrophils. The migration and inflammation targeting ability of neutrophils were not affected by the captured NPNs, as evidenced by the similar ratios of neutrophils/tumor cells in PTT-treated tumors compared with tumors of mice treated with PEG-*b*-PLGA NPs or phosphate-buffered saline (PBS) (Supplementary Fig. 18). Moreover, serum levels of proinflammatory cytokines interleukin (IL)-6 and tumor necrosis factor (TNF)-α were significantly increased at 4 h post injection of NPNs but dropped to normal levels at 24 h (Supplementary Fig. 19), indicating that NPNs only induced a transient inflammation.

**NPNs accumulated in PTT-treated tumors along with neutrophil recruitment.** To verify whether NPNs could be delivered to tumors via the chemotaxis of neutrophils, we assessed the percentage of neutrophils that contained NPNs in the tumor. The flow cytometric results revealed that, in the tumor, about 19% of neutrophils contained NPNs, while there were only 7% of neutrophils that internalized NPs (Fig. 5h and Supplementary Fig. 20). In addition, the amount of cellular NPNs taken up by neutrophils was higher than that of NPs (Fig. 5i). Besides neutrophils, most nanoparticles were predominately uptaken by macrophages rather than other immune cells (Supplementary Fig. 21).

To monitor the accumulation of NPNs in the entire inflamed tumor tissue using IVIS, DiD-labeled NPNs were *i.v.* injected into untreated or PTT-treated EMT6-bearing mice. The results showed that more NPNs targeted to PTT-treated tumors than to untreated tumors (Fig. 5j and Supplementary Fig. 22), as further confirmed by a quantitative ROI analysis of DiD fluorescent signals in the tumor areas (Fig. 5k). At the end of the experiment, we harvested the tumors and major organs, and analyzed their DiD fluorescent signals using IVIS. In accordance with the results in Fig. 5j, k, more DiD-labeled NPNs accumulated in PTT-treated tumors than in untreated tumors (Fig. 5l and Supplementary Fig. 23), as further substantiated by a quantitative ROI analysis (Fig. 5m). Moreover, the tumor targeting ability of the neutrophil-hitchhiking NPNs after PTT was even higher than that of EPR effect-based PEG-*b*-PLGA NPs (Supplementary Fig. 24). Taken together, these results indicated that more pathogen-mimicking NPNs could be taken up by

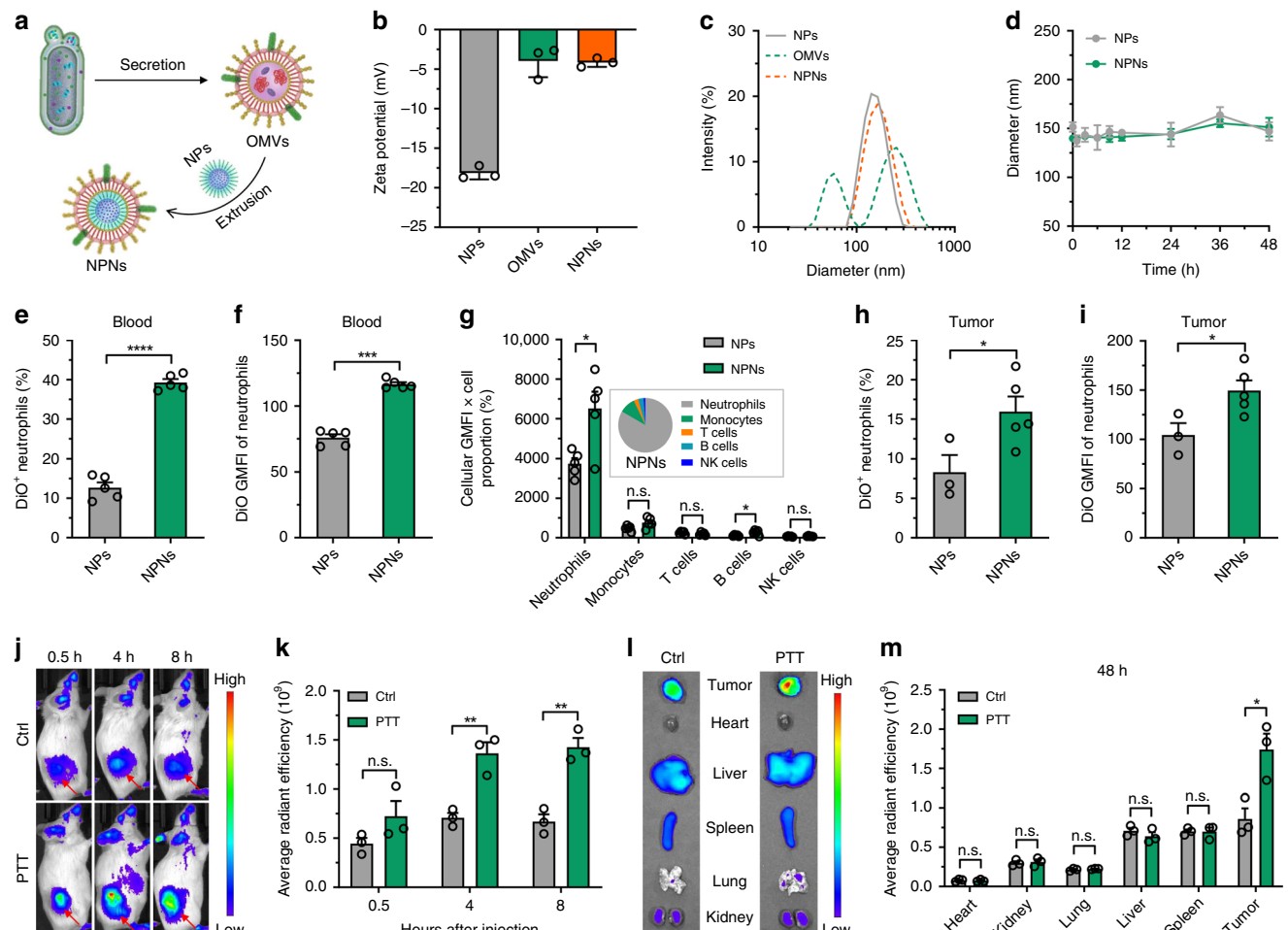

**Fig. 5 Pathogen-mimicking NPNs hitchhiked circulating neutrophils and homed to inflamed tumors. a** Schematic showing the preparation of NPNs.
**b** Zeta potential and **c** size distribution of NPNs before and after coating with OMVs. $n = 3$ biologically independent samples per group. **d** Stability of NPNs
compared to PEG-*b*-PLGA NPs in medium containing 10% mouse serum. $n = 3$ per group. **e–i** DiO-labeled NPs and NPNs were *i.v* injected into PTT-treated
EMT6-bearing mice. After 4 h, the percentage of neutrophils with NPs in the blood and tumor were analyzed by flow cytometry. **e** Percentage of DiO⁺
neutrophils and **f** DiO GMFI of neutrophils in blood were analyzed by flow cytometry. $n = 5$ per group. **g** Distribution of DiO-labeled NPs and NPNs in
different immune cells in blood as defined by multiplying the percentage of immune cells with cellular GMFI of DiO. The inset showed the relative
distribution of NPNs in the major immune cell types in blood as calculated using the following equation $(GMFI_x \times percentage_x)/\Sigma(GMFI_x \times percentage_x)$, "$x$"
represents neutrophils, monocytes, T cells, B cells, or NK cells. $n = 5$ per group. **h** Percentage of DiO⁺ neutrophils and **i** DiO GMFI of neutrophils in tumors
were analyzed by flow cytometry. $n = 3$ for the NPs group and $n = 5$ for the NPNs group. **j–m** DiD-labeled NPNs were *i.v.* injected into control or PTT-
treated EMT6-bearing mice. $n = 3$ per group. **j** At different time points, in vivo DiD fluorescent signals were observed with IVIS. **k** Quantitative ROI analysis
of DiD fluorescent signals in tumor areas. **l** At 48 h after injection, the tumors and major organs were collected and fluorescent images were acquired with
IVIS. **m** Quantitative ROI analyses of DiD signals in tumors and organs. Data are shown as mean ± SD (**b, d**) or mean ± SEM (**e–i, k, m**), and analyzed by
unpaired two-tailed Student's $t$-test. *$P < 0.05$, **$P < 0.01$, ***$P < 0.001$, ****$P < 0.0001$. n.s. not significant. Source data are provided as a Source Data file.

neutrophils in blood and then efficiently transported to PTT-
treated tumors along with neutrophil recruitment.

**Transfer of loaded cargos from neutrophils to tumor cells
under inflammatory conditions.** Next, we assessed whether the
encapsulated cargos could be released from neutrophils and
transferred to tumor cells under PTT-caused inflammatory con-
ditions. A medium containing 100 nM of phorbol myristate
acetate (PMA) was used to mimic the inflammatory conditions[38].
PMA-treated neutrophils showed a clear formation of NETs,
which manifests the disintegration of the neutrophil plasma
membrane and may give rise to the release of encapsulated cargos
(Fig. 6a and Supplementary Fig. 25). DiD-labeled neutrophils
were incubated with DiO-labeled PEG-*b*-PLGA NPs for 1 h; the
dual-labeling efficiency was ~100% (Supplementary Fig. 26). The

obtained neutrophils were co-cultured with EMT6 cells for 5 h in
the presence of 100 nM of PMA (Fig. 6b). Microscopy observa-
tion revealed that the number of EMT6 cells internalized NPs
under inflammatory condition was about four times more than
that under normal condition (Fig. 6c, d). The cytotoxicity of
NPNs@Pt released from PMA-treated neutrophils against EMT6
cells was comparable to that of naïve NPNs@Pt, indicating that
the NPs released by PMA-treated neutrophils remained inte-
grated and active (Supplementary Fig. 27).

We further verified this phenomenon in vivo. First, in vivo
NETosis in PTT-treated tumor was verified by the co-localization
of Ly6G and citrullinated histone H3 (Cit-histone H3) (Supple-
mentary Fig. 28), a widely accepted marker of NETs[39]. Next,
DiD-labeled neutrophils encapsulating NPs@DiO were intratu-
morally injected into PTT-treated EMT6 tumors. After 20 h, the
tumors were excised for analysis of DiO fluorescent signals in

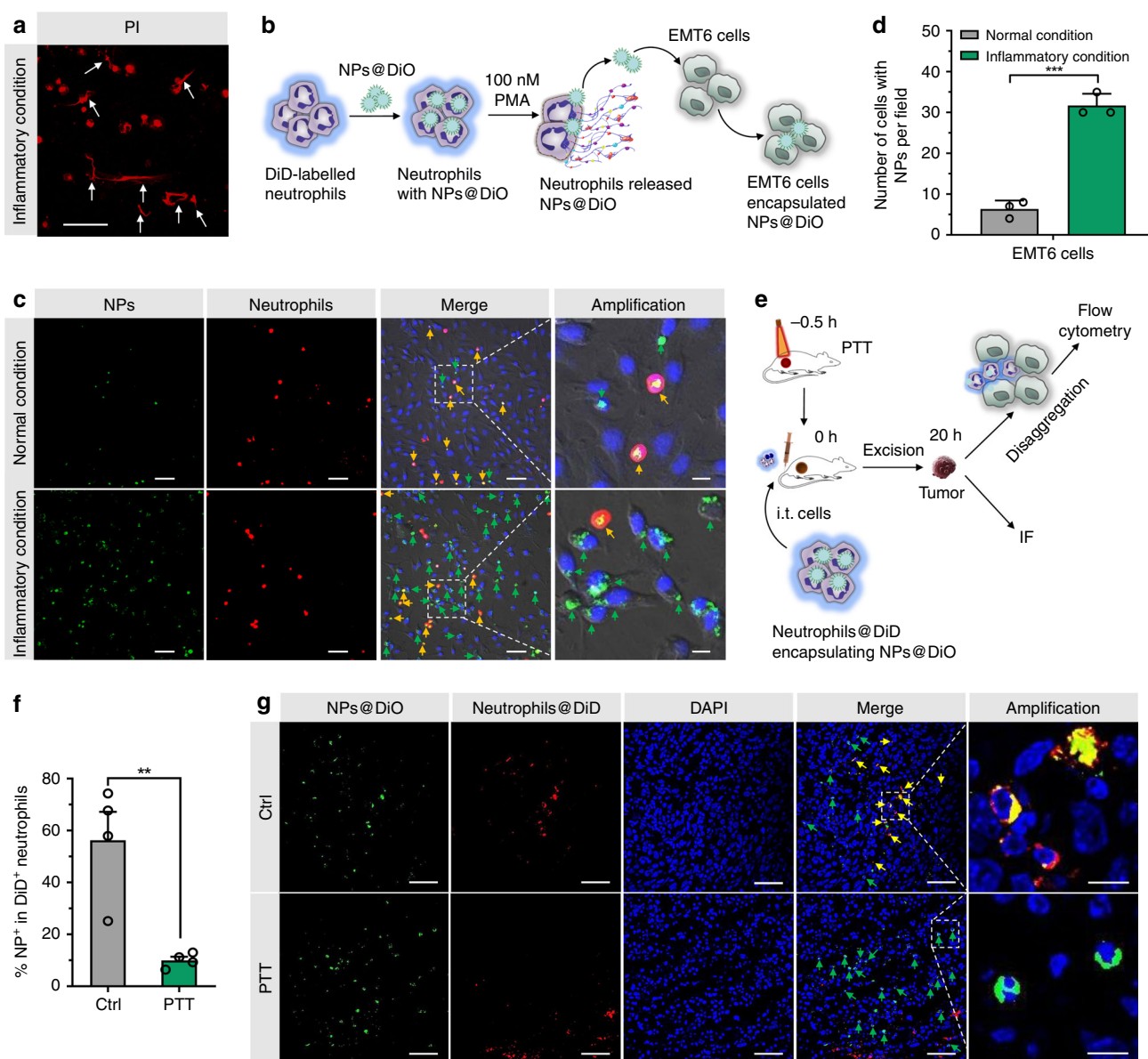

**Fig. 6 Tumor cells internalized NPs that were released from neutrophils under inflammatory conditions. a** Confocal fluorescent images of propidium iodide (PI)-stained neutrophils cultured under inflammatory conditions (100 nM PMA) for 5 h. Scale bar, 50 μm. Data are representative of two independent experiments. **b** Workflow of neutrophils and EMT6 co-culture assay. DiD-labeled neutrophils were incubated with DiO-labeled PEG-*b*-PLGA NPs (NPs@DiO) for 1 h. The obtained neutrophils were co-cultured with EMT6 cells followed by nucleus staining and microscopy imaging. **c** Confocal fluorescent images of co-cultured EMT6 cells and neutrophils. Yellow arrows indicated neutrophils with NPs and green arrows indicated EMT6 cells with NPs. Scale bars, 10 μm for the amplification panels, and 50 μm for all the other panels. Data are representative of three biological replicates. **d** Number of EMT6 cells that internalized NPs under inflammatory and normal conditions. $n = 3$ biologically independent samples per group. **e** Schematic showing analyses of tumors that internalized NPs released from neutrophils in untreated or photothermally treated EMT6 tumors. EMT6-bearing mice were left untreated or were *i.v.* injected with NPs@PBT on −12 h and tumors were performed with PTT (40 °C for 5 min) on −0.5 h. DiD-labeled neutrophils were incubated with DiO-labeled PEG-*b*-PLGA NPs and intratumorally (i.t.) injected into the mice at 0 h. After 20 h, DiO fluorescent signals were analyzed by flow cytometry and IF. **f** The percentage of transferred neutrophils containing NPs (defined as NP$^+$DiD$^+$) was detected by flow cytometry. $n = 4$ mice per group. **g** DiO signals in transferred neutrophils and EMT6 tumor cells were analyzed by IF. Yellow and green arrows indicated NPs@DiO that were co-localized and not co-localized with neutrophils, respectively. Scale bars, 10 μm for the amplification panels, and 50 μm for all other panels. Data are representative of three biological replicates. Data are shown as mean ± SD (**d**) or mean ± SEM (**f**), and analyzed by unpaired two-tailed Student's *t*-test. **P < 0.01, ***P < 0.001. Source data are provided as a Source Data file.

transferred neutrophils and EMT6 tumor cells by flow cytometry and IF (Fig. 6e). As shown in Fig. 6f and Supplementary Fig. 29a, in PTT-treated tumors, the percentage of transferred neutrophils containing NPs@DiO decreased by 77.6% compared with the control group, proving that PTT induced the efficient release of cargos from neutrophils. The results of IF also showed that in PTT-treated tumors, fewer NPs@DiO signals were co-localized with neutrophils (Fig. 6g). In contrast, more NPs@DiO were taken up by EMT6 tumor cells after PTT from the flow cytometric results (Supplementary Fig. 29b, c). This was further confirmed by the high co-localization of NPNs@DiD with EGFP expressing EMT6 tumor cells (EMT6-EGFP) in tumor slices

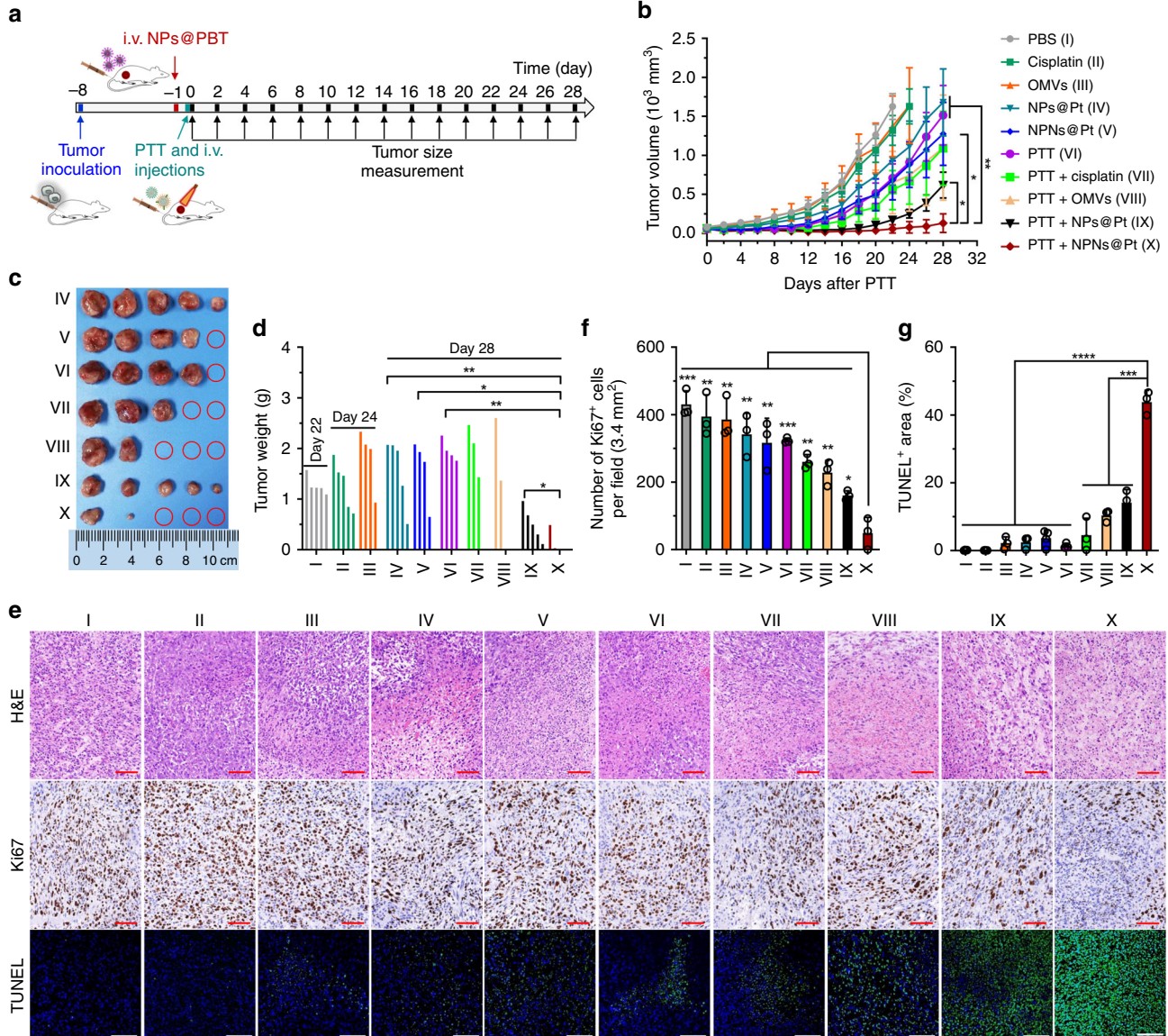

**Fig. 7 PTT in combination with NPNs@Pt enhanced anti-tumor effect. a** Workflow for treatment of EMT6-bearing mice using PTT-based neutrophil-mediated NPNs@Pt delivery system. On day -1 (defined as the 7th day after tumor inoculation), mice were *i.v.* injected with NPs@PBT. On day 0, the mice were received PTT treatment and were administrated with PBS, cisplatin, OMVs, NPs@Pt, or NPNs@Pt after 30 min. Tumor volumes were measured every two days from day 0 to day 28. **b** Tumor growth curves during the treatments. **c** Photos and **d** weight of the tumors collected on day 28. $n = 5$ mice per group for **b**–**d**. **e** Representative images of H&E, Ki67, and TUNEL staining of tumor tissues obtained on day 3 after the PTT treatment. Scale bars, 100 μm. **f** Number of Ki67$^+$ cells per field (3.4 mm$^2$) in the tumor slices in **e**. Significant analyses were performed compared with Group X. $n = 3$ per group. **g** Percentage of TUNEL$^+$ area in the tumor slices in panel **e**. $n = 4$ for group I, II, V, and $n = 3$ for all other groups. Data are shown as mean ± SEM (**b**) or mean ± SD (**f**, **g**), and analyzed by unpaired two-tailed Student's *t*-test. *$P < 0.05$, **$P < 0.01$, ***$P < 0.001$, ****$P < 0.0001$. Source data are provided as a Source Data file.

(Supplementary Fig. 30). These results indicate that under PTT-induced inflammatory conditions, neutrophils released encapsulated NPs upon membrane disruption driven by NETs formation, which were then endocytosed by the EMT6 tumor cells.

**A single treatment of PTT plus NPNs@Pt significantly suppressed tumor growth.** Finally, we studied the anti-tumor effect of PTT in combination with neutrophil-mediated chemotherapy using EMT6-bearing mice. On day 0 (defined as the 8th day after tumor inoculation), the mice received PTT treatment (40~41 °C for 5 min, Supplementary Fig. 31) on the tumors followed by *i.v.* injection of different formulations (Fig. 7a). As shown in Fig. 7b,

compared with the PBS-treated group, the cisplatin- and OMVs-treated groups showed ~17 and 16% inhibition of tumor growth, which might be caused by the cytotoxicity of cisplatin to tumor cells and an immune response induced by OMVs, respectively[40]. Compared with cisplatin treatment, NPs@Pt treatment resulted in ~32% inhibition of tumor growth, probably due to their enhanced tumor accumulation resulted from the EPR effect. NPNs@Pt resulted in ~24% extra tumor inhibition compared to NPs@Pt, which might be the result of a combined effect of increased neutrophil carriage-induced accumulation in the tumor and OMVs-initiated immune response. Treatment using PTT in combination with cisplatin or OMVs showed ~28% and ~27% extra tumor inhibition compared with PTT alone. Treatment with

PTT in combination with NPs@Pt showed ~43% tumor inhibition compared with PTT in combination with cisplatin, again indicating the advantage of nanomedicines in cancer drug delivery. Obviously, treatment with PTT combined with NPNs@Pt showed the best therapeutic effect, obtaining ~79% tumor inhibition as compared to treatment with PTT combined with NPs@Pt. This may primarily be due to more cisplatin in the tumor carried by abundant neutrophils after PTT. On days 22 and 24, PBS-, cisplatin- and OMVs-treated mice were euthanized when the tumor volume reached 2000 mm³. On day 28, the tumor tissues of all other groups were harvested, weighed and photographed. Weights and photos of tumors further proved excellent anti-tumor effect of treatment with PTT in combination with NPNs@Pt (Fig. 7c, d). In addition, hematoxylin and eosin (H&E), Ki67, and terminal deoxynucleotidyl transferase-mediated dUTP nick end-labeling (TUNEL) staining showed that tumors treated with PTT in combination with NPNs@Pt had substantial apoptosis and decreased proliferation (Fig. 7e–g). The results regarding body weight showed that OMVs treatment triggered a transient weight loss, indicating some acute response caused by OMVs (Supplementary Fig. 32).

We next validated that the superior anti-tumor therapeutic outcome of the combination therapy was attributed to the neutrophil-mediated drug delivery. In mice bearing CT26 tumors with low-tumor neutrophil filtration, PTT + NPNs@Pt did not display better anti-tumor effects than PTT+ NPs@Pt, suggesting the necessity of neutrophils as drug carriers (Supplementary Fig. 33). To further prove this, EMT6-bearing mice were received an intraperitoneal (*i.p.*) injection of α-Ly6G (200 μg) before the therapy to deplete neutrophils (Supplementary Fig. 34a, b). As a consequence, the absence of neutrophils significantly impaired the efficacy of the combined therapy and decreased the mouse survival (Supplementary Fig. 34). We next studied the role of NETosis followed by cargo release from neutrophils in the therapeutic effects. A protein arginine deiminase (PAD) inhibitor BB-Cl-amidine was *i.p.* injected into mice to inhibit NETosis, which also compromised the therapeutic outcomes of NPNs@Pt-based combination therapy and reduced the mouse survival as defined by the Kaplan-Meier estimator (Supplementary Fig. 35). These results collectively confirmed the vital role of neutrophils in mediating tumor-specific delivery and release of NPNs@Pt in response to PTT.

**Two treatments of PTT plus NPNs@Pt completely eradicated the tumors**. We further investigated the tumor therapeutic effects on the tumor-bearing mice that received two treatments of PTT plus NPNs@Pt. On days 0 (tumor volumes ~80 mm³) and 3 (defined as the 8th and 11th day after tumor inoculation), the mice received two indicated treatments (Fig. 8a). Compared with the PBS group, NPs@Pt treatment resulted in ~26% inhibition of tumor growth, demonstrating the anti-tumor effect of nanomedicines (Fig. 8b–h). In comparison with NPs@Pt, NPNs@Pt obtained ~52% inhibition of tumor growth with 20% of mice free of tumor, probably due to the integrated effect of more NPNs@Pt accumulation in the tumor mediated by neutrophil delivery and OMVs-initiated anti-tumor immune response. Treatment with PTT in combination with NPs@Pt showed ~71% tumor growth inhibition compared with NPs@Pt treatment alone. Additionally, 40% of the treated mice were free of tumor, proving the enhanced anti-tumor effect of PTT combined with chemotherapy. Strikingly, treatment with PTT in combination with NPNs@Pt resulted in an impressive rate of EMT6 tumor rejection with 100% mice free of tumors, probably resulting from the eradication of residual tumor cells mediated by the enhanced accumulation of NPNs@Pt. Administration of OMVs caused a transient

drop in the mouse weight, which, however, was recovered within 5 days after the last treatment (Fig. 8i). On the 20th day after the first treatment, the tumor tissues were harvested, weighed, and photographed, which further confirmed the superior anti-tumor effect of treatment with PTT in combination with NPNs@Pt (Fig. 8j–l). Since the dosage and frequency of cisplatin were low, the treatments did not induce obvious side effects to tissues of spleen, liver and lung from histological examination (Supplementary Fig. 36).

## Discussion

In the present study, we developed a pathogen-mimicking NPN system to in situ hitchhike circulating neutrophils for the complete eradication of residual tumors after PTT. Specifically, our approach includes the following consecutive steps (Fig. 1). (1) PTT creates an inflammatory environment in tumors, which produces chemokines to stimulate the production and activation of neutrophils and broadcasts the tumor locations to the neutrophils. (2) The pathogen-mimicking NPNs hitchhike the circulating neutrophils through the recognition of PAMPs on NPNs by pattern recognition receptors (PRR) on neutrophils. (3) Neutrophils laden with NPNs actively home to tumor sites, extravasate the endothelium lining vasculature, and penetrate the tumor microspace, thus overcoming biological barriers encountered by a conventional drug delivery approach. (4) Once transported into tumor tissues, NPNs were released from neutrophils upon the disruption of the plasma membrane during the NETs formation, and then were internalized by tumor cells to exert anti-tumor effects.

A number of studies have reported the successful use of phototherapy, such as PTT and photodynamic therapy (PDT), in the treatment of cancers[41–46]. In our study, PTT only moderately delayed the tumor growth. In general, the limited light penetration in PTT causes heterogeneous heat distribution within the tumor tissue and fails to completely remove the whole tumor, thus leading to lethal tumor recurrence and post-therapy metastasis. Although the therapeutic efficacy can be improved, for example, by increasing the laser power or irradiation time, PTT involving a high temperature still has the drawbacks of overheating and skin burning[47].

We have proved that PTT created an inflammatory tumor microenvironment regardless of the tumor type and tumor size. The concentration gradient of chemokines such as CXCL1 and MIP-2 broadcasted to neutrophils the location of residual tumors that PTT alone failed to remove. Live neutrophils have been reported to efficiently target primary tumors[4,18,19]. In our work, we found that such tumor accumulation capability of neutrophils could be improved by ~300 to 600% via PTT pretreatment of the tumors. Such an elevated accumulation of neutrophils could be attributed to their capacity to actively home to inflammatory sites, extravasate from the vasculature, and transmigrate deep to non-vascular regions along with the chemotaxis gradient. In addition, we observed in real-time the complete process of neutrophil recruitment from the blood to the tumor interstitial stroma after PTT via live-imaging (Fig. 4c–e). A worthy concern is whether neutrophils could efficiently infiltrate the tumor region with low perfusion blood vessels (e.g., in the core area of solid tumors). We postulated that neutrophils could still actively migrated into those areas following the concentration gradient of chemokines based on previous reports[48,49].

The pathogen-mimicking concept was derived from the manner in which neutrophils in nature fight invading pathogens[50]. In this work, we used OMVs to coat NPs because OMVs inherit the PAMPs population from their parent bacteria while avoiding toxicity issues from the use of live bacteria[37,51,52]. In the PTT-

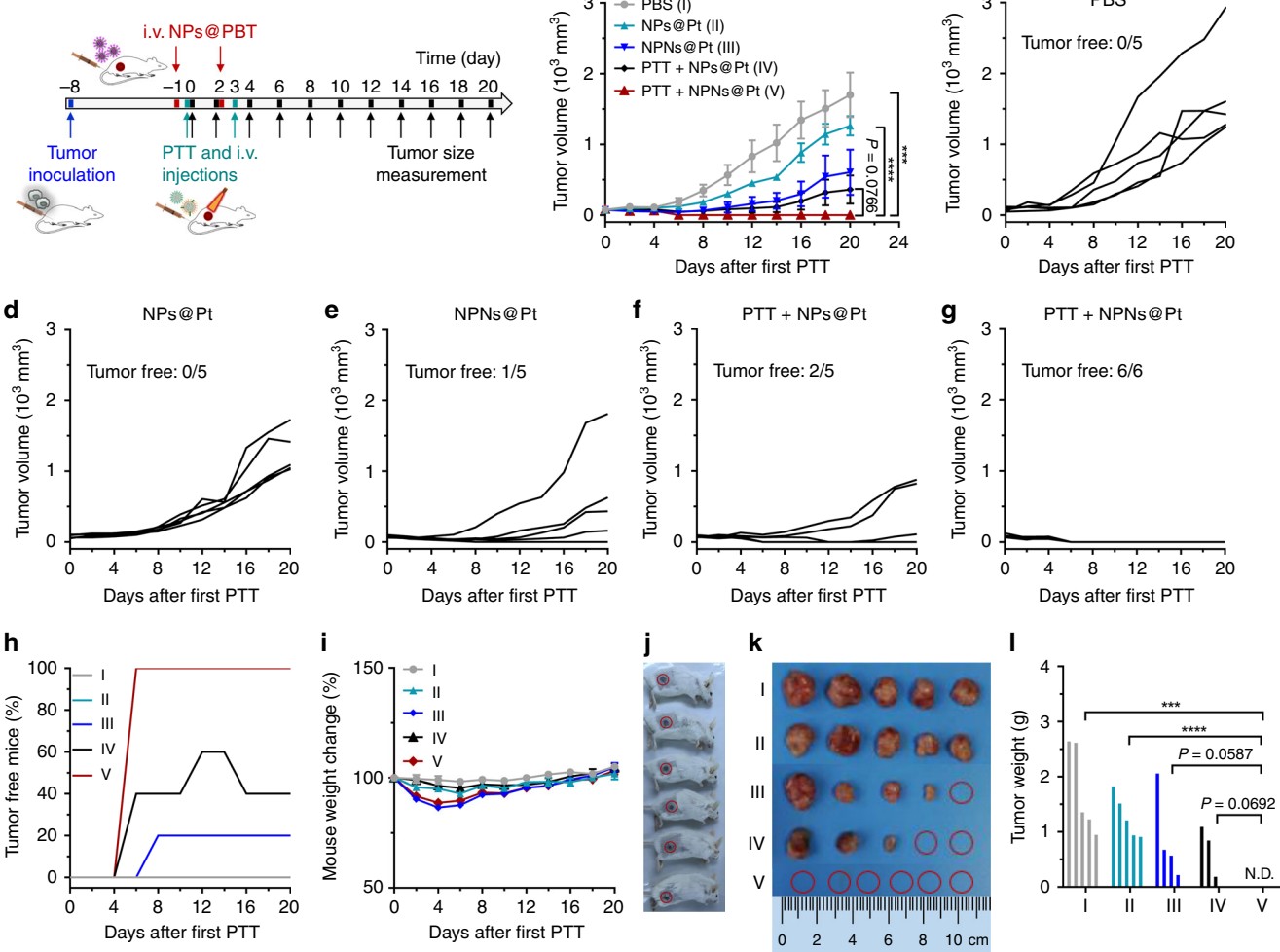

**Fig. 8 Two treatments of PTT plus NPNs@Pt completely eradicated tumors in all mice. a** Workflow for treatment of EMT6-bearing mice using PTT-based neutrophil-mediated NPNs@Pt delivery system. On day -1 and day 2 (defined as the 7th and 10th day after tumor inoculation, respectively), mice were *i.v.* injected with NPs@PBT. On days 0 and 3, the mice received PTT treatment on the tumors followed by an *i.v.* injection of PBS, NPs@Pt, or NPNs@Pt. Tumor volumes were measured every two days from day 0 to day 20. $n = 6$ for PTT + NPNs@Pt, and $n = 5$ for the other groups. **b** Tumor growth curves during the treatments. **c–g** Growth of individual tumors during the treatments. **h** The percentage of tumor-free mice during the treatments. **i** Change of mouse weight during the treatments. **j** Photos of mice treated with PTT + NPNs@Pt on day 20, which showed complete eradication of the EMT6 tumors in all six mice. **k** Photos and **l** weight of the tumors collected on day 20. Data are shown as mean ± SEM and analyzed by unpaired two-tailed Student's *t*-test. ***$P < 0.001$, ****$P < 0.0001$. N.D. not detected. Source data are provided as a Source Data file.

induced acute inflamed tumor model, the neutrophil targeting efficacies in the peripheral blood were ~41% (Fig. 5e), even higher than that using CD11b-decorated NPs (~30%)[19]. Moreover, neutrophils internalized 83.4% of the total NPNs, compared to monocytes and other types of cells (Fig. 5g). In addition, compared to the reported antibody decoration method, the present approach is likely more generalizable and can be easily applied to modify other particular systems.

After therapeutic agents are delivered to the tumor site, their release from cell carriers and subsequent internalization by cancer cells are necessary to exert the anti-tumor effect. Previously developed cell carriers, such as red blood cells (RBCs), monocytes, macrophages, and stem cells, release NPs through passive diffusion, exocytosis, and efflux[14]. The passive and slow release of NPs might cause their intracellular degradation before being released. In contrast, neutrophils can rapidly form NETs and cause the disintegration of cell membrane within 4 h in response to inflammatory stimuli[4,53], which triggers a fast and specific release of the cargos. In our studies, we observed the release of NPNs from neutrophils in response to inflammation from both

the in vitro co-culture model and the in vivo PTT-treated tumor model (Fig. 6c–g).

We observed dramatic therapeutic outcomes using such an NPN system. A single treatment of PTT in combination with NPNs@Pt resulted in a 60% tumor-free rate and a 97% tumor growth inhibition (Fig. 7b, c). More importantly, repeated combination treatments completely eradicated all of the tumors in each of the treated mice on two occasions (Fig. 8). It is worth noting that we observed a transient weight loss during the OMV treatments (Supplementary Fig. 32 and Fig. 8i), possibly due to the acute enteritis induced by PAMPs on OMVs (such as LPS)[54]. Such OMVs toxicity issues can be addressed by using bacteria strains with lower pathogenicity or genetically modified attenuated bacteria strains with, for example, impaired LPS signaling[40,55].

In conclusion, we established a NPN system that can in situ hitchhike circulating neutrophils to overcome obstacles encountered by NPs. The NPNs containing PAMPs can be recognized by PRRs on neutrophils and be subsequently encapsulated. Neutrophils loaded with NPNs actively migrated to tumor stroma by

crossing biological barriers and then released NPNs in response to inflammatory stimuli. The released NPNs were subsequently internalized by tumor cells and functioned intracellularly to kill them. Most importantly, the combined therapy of PTT and NPNs@Pt completely eradicated tumors in all the treated mice. In total, our work offers a method to in situ hitchhike circulating neutrophils with pathogen-mimicking NPNs for improved nanomedicine delivery and tumor therapy.

## Methods

**Materials**. PEG-*b*-PLGA was synthesized by ring-opening polymerization according to the literature[56]. Typically, a mPEG (5.0 g) and the 3:1 molar ratio of lactide (9.53 g)/glycolide (2.56 g) were added into a flask. The catalyst Sn(Oct)$_2$ (2 drops) was then added with 1:1 molar ratio with mPEG and the reaction was allowed to proceed at 125 °C. After 2 h, the product was dissolved in CHCl$_3$ and precipited in ether three times. Finally, the product was dried by vacuum.

GNRs were synthesized according to previous reports[57]. First, preparation of seed solution. 500 μL of 10 mM HAuCl$_4$ was add to 15 mL of 100 mM hexadecyltrimethylammonium bromide (CTAB) solution in glass vial with small stir bar. Then 1.2 mL 10 mM NaBH$_4$ solution was added to the mixture. Keep the seed solution in dark and use it within 2 h of synthesis. Second, synthesis of nanorods. 1.7 mL of 10 mM HAuCl$_4$, 250 μL of 10 mM AgNO$_3$, and 270 μL of 100 mM Ascorbic acid were added to 40 mL of 100 mM CTAB in 100 mL round bottom flask with stir bar. Shake until solution turns clear. Add 840 μL of seed solution to the above solution, and let sit still at room temperature for 40 min. Nanorods were washed by PEG-LA solution to remove free CTAB by centrifugation.

Cisplatin was purchased from Shandong Boyuan Pharmaceutical (Jinan, China). Pt (IV) prodrug was prepared according to a previous report[58]. First, dry cisplatin (0.50 g, 1.65 mmol) was suspended in water (10 mL) and H$_2$O$_2$ (30% w/v, 10.0 mL) was added. The mixture was stirred for 4 h at 50 °C and a pale yellow powder was obtained. Recrystallization of cis,trans,cis-[PtCl$_2$(OH)$_2$(NH$_3$)$_2$] was performed in situ, collected and washed with cold water, ethanol and ether, and dried in a desiccator. Second, synthesis of c,c,t-[Pt(NH$_3$)$_2$Cl$_2$(O$_2$CCH$_2$CH$_2$CH$_2$CH$_2$CH$_2$CH$_2$CH$_2$CH$_2$CH$_3$)$_2$]. Decanoic anhydride (1.21 g, 3.71 mmol) was added to a solution of c,c,t- [Pt (NH$_3$)$_2$Cl$_2$(OH)$_2$] (0.25 g, 0.74 mmol) in DMSO (8 mL), and the reaction mixture was stirred at room temperature for 72 h. Water was added to the mixture to precipitate a light yellow solid, which was then dissolved in acetonitrile. A yellow solid (Pt(IV) prodrug) was obtained after rotary evaporation of the acetonitrile solution, and then was washed several times with diethyl ether and dried.

DiO and DiD were purchased from Biotium (Fremont, CA, USA). 4′,6-Diamidino-2-phenylindole (DAPI) and PMA were purchased from Sigma-Aldrich (St. Louis, MO, USA). PI and all antibodies for flow cytometry were purchased from Biolegend (San Diego, CA, USA). Other chemicals and reagents were all of analytical grades and used as received without otherwise statement.

**Cell lines and mice**. Murine breast cancer cell line EMT6 and colon adenocarcinoma cell line CT26 were obtained from American Type Culture Collection (ATCC, Manassas, VA, USA). EMT6-EGFP cells were constructed to express EGFP in EMT6 cells. EMT6 and EMT6-EGFP cells were cultured in Dulbecco's Modified Eagle Medium (DMEM, Gibco, Waltham, MA, USA) containing 10% (v/ v) fetal bovine serum (FBS, Excell Bio, Shanghai, China). CT26 cells were cultured in RPMI-1640 Medium (Gibco) containing 10% (v/v) FBS.

Female BALB/c mice and male C57BL/6 mice (6–8 weeks) were purchased from Beijing Vital River Laboratory Animal Technology (Beijing, China). $Tlr4^{-/-}$ mice on C57BL/6 background were bred in SPF conditions. All mice used in this study were housed in a specific pathogen-free facility, received human care, and were used according to the animal care regulations of University of Science and Technology of China.

**Tumor inoculation**. The subcutaneous breast and colon tumor models were established by subcutaneously injecting EMT6 (5.0 × 10$^5$ cells),; EMT-EGFP (5.0 × 10$^5$ cells), or CT26 (1.0 × 10$^6$ cells) cells suspended in 50 μL PBS (0.01 M, pH 7.4) into the right flank of BALB/c mice.

**NP preparation**. PBIBDF-BT was synthesized according to previous report[28]. Tri (o-tolyl)phosphine (P(o-tol)$_3$, 0.008 g, 0.025 mmol), tris(dibenzylideneacetone) dipalladium (Pd$_2$(dba)$_3$, 0.006 g, 0.0082 mmol), were added to a solution of (3E,7E)-3,7-bis(6-bromo-1-(2-decyltetradecyl)-2-oxoindolin-3-ylidene)benzo[1,2-b:4,5-b′]difuran-2,6(3H,7H)-dione (0.2 g, 0.16 mmol) and 5,5′-bis(tributylstannyl)-3,3′-bis(dodecyl)-2,2′-bithiophene (0.13 g, 0.16 mmol) in toluene (6 mL) under nitrogen. After subjecting to three cycles of evacuation and admission of nitrogen, the mixture was then heated to 110 °C for 48 h. After cooling to room temperature, the mixture was added into methanol and then stirred for 2 h. A black precipitate was obtained after filtration. After washing with methanol and petroleum ether and removing solvent, the black solid product was obtained. To prepare NPs@PBT, a 500 μL CHCl$_3$ solution containing 10 mg PEG-*b*-PLGA, 1 mg PBIBDF-BT, and 2 mL ultrapure water were emulsified by sonication at 104 W for 2 min over an ice bath using a microtip probe sonicator (JY92-IIN, Scientz Biotechnology, Ningbo, China). Then the mixture was stirred under reduced pressure for 30 min to remove CHCl$_3$ and then concentrated by ultrafiltration (molecular weight cutoff = 100 KDa, Millipore, Burlington, MA, USA) and were stored at 4 °C for further use.

NPs@DiO, NPs@DiD, and NPs@Pt were prepared using a dialysis method. Briefly, 10 mg PEG-*b*-PLGA, 25 μg DiO or DiD, or 2 mg Pt(IV) prodrug, were dissolved in DMSO and stirred in a round-bottomed flask for 1 min at room temperature. Then, five volumes of Milli-Q ultrapure water (Millipore, Bedford, MA, USA) were added into the flask under vigorous stirring. After stirring for another 10 min at room temperature, the solution was transferred into a dialysis bag (Spectra/Por®, Float-A-Lyzer, molecular cutoff = 14 KDa, Spectrum, Waltham, MA, USA) and dialyzed against ultrapure water overnight to remove dimethyl sulfoxide (DMSO). Then, the resulting NPs were concentrated by ultrafiltration (Millipore) and were stored at 4 °C for further use.

**Photothermal effect of NPs@PBT under 808 nm laser**. The aqueous solutions of NPs@PBT (25 μg mL$^{-1}$ PBIBDF-BT) were irradiated with an 808 nm laser (1.5 W cm$^{-2}$) (New Industries Optoelectronics, Changchun, China) connected to a square waveform generator (10 Hz), which controls the power delivered to the laser. The duty cycle for a square waveform was set as 65%, 75%, and 90% ON, respectively. The temperature changes of NPs@PBT suspensions under the laser irradiation were detected by using an infrared camera (ICI7320, Infrared Camera Inc., TX, USA).

**Preparation of bacterial OMVs**. Bacterial OMVs from *E. coli* (strain Trans T1) were prepared according to the literature[24,40]. *E. coli* was inoculated into Luria broth (LB) medium and cultured with shaking (200 rpm) at 37 °C for about 12 h. Then the bacteria suspension were added to fresh LB medium at a 1:100 dilution and cultured for another 4 h until the OD$_{600}$ value of the bacterial suspension reached ~1.0. Next, bacterial culture was centrifuged at 4000 × *g* for 10 min at 4 °C to remove the bacteria and the supernatant was collected and filtered with a 0.45 μm vacuum filter. Then the filtrate was concentrated by ultrafiltration (100 KDa, Millipore). The concentrated solution was then pelleted by centrifuging with an SW 70 Ti rotor (Beckman coulter, Atlanta, Georgia, USA) at 150,000 × *g* for 2 h at 4 °C. The obtained OMVs were dissolved in water and stored at −80 °C for further use.

**Preparation of NPNs**. The obtained OMVs were extruded 21 times through a 200 nm polycarbonate membranes (Whatman, Chicago, IL, USA). To prepare NPNs, the 0.2 mg extruded OMVs and 1 mg NPs@DiO, NPs@DiD or NPs@Pt were mixed and extruded another 21 times through a 200 nm polycarbonate membrane.

**Characterization of NPs and NPNs**. The size and zeta potential of NPs@PBT, NPs, OMVs, and NPNs were detected by a Malvern Zetasizer Nano ZS90 dynamic light scattering (DLS) instrument (Malvern Instruments Ltd., Worcs, UK) at 25 °C. To assess the stability of NPs and NPNs, these particles were incubated in 10% mouse serum at a concentration of 0.5 mg mL$^{-1}$ and the diameters of particles at indicated time points were detected by DLS at 25 °C in triplicate. The morphology of NPNs, NPs@PBT, and GNRs were characterized by TEM (JEM-2011, Tokyo, Japan). The absorbance spectra of NPs@PBT and GNRs were assessed using an ultraviolet spectrophotometer (Cary 60, Agilent technologies, California, USA).

**PTT of tumor**. For PTT treatment of tumor, 12 h after *i.v.* injection of NPs@PBT or GNRs, the tumor was irradiated with an 808 nm laser and the temperature of tumor site was monitored with an infrared camera. The tumor temperature was kept at 40, 42 and 45 °C for 2, 5, and 10 min by carefully regulating the frequency of laser with signal generator, respectively. For flow cytometry and enzyme-linked immunosorbent assay (ELISA), blood and tumors were harvested at different time points after PTT. For IF, tumors were harvested 4 h after PTT. For live-imaging, tumors were observed with a confocal microscopy after PTT. To analyze whether CXCR2 signaling mediated the chemotaxis of neutrophils to PTT-treated tumors, CXCR2 inhibitor SB225002 (Med Chem Express, New Jersy, USA) was *i.p.* injected into mice at a dose of 10 mg per kg 30 min before PTT treatment.

**Isolation and analyses of leukocytes in blood and tumor**. Blood leukocytes were obtained by lysing erythrocytes and centrifugation. To obtain tumor infiltrating leukocytes, tumor tissues were harvested, excised into small pieces and suspended in 5 mL digestion solution (1 mg mL$^{-1}$ type IV collagenase [Sigma-Aldrich] in RPMI-1640 medium containing 10% FBS) and incubated at 37 °C for 1.5 h with persistent agitation. Digested cells were filtered by a 200-mesh nylon sieve and then collected by centrifugation at 652 × *g* for 5 min at 4 °C. Then the cells were resuspended in 3 mL 40% percoll (GE Healthcare, Chicago, IL, USA) and collected by centrifugation at 805 × *g* for 30 min at 25 °C. Tumor-infiltrating leukocytes were obtained after lysing erythrocytes.

**Isolation of neutrophils from bone marrow**. Neutrophil isolation was performed according to a previous method[4]. Briefly, bone marrow cells were flushed from murine femurs and tibias. After lysing erythrocytes, cells were resuspended in PBS,

added onto a percoll mixture consisting of 55, 65, and 78% percoll (v:v) and centrifuged at $805 \times g$ for 30 min. Neutrophils were collected from the interface of 65 and 78% percoll layers and washed twice with ice-cold PBS. The purity of neutrophils was analyzed with flow cytometry.

**Flow cytometry.** Cells in a single-cell suspension were incubated with mouse CD16/32 antibody to saturate Fc receptors before staining with the indicated fluorescently labeled antibodies for surface antigens. We performed flow cytometry on a BD FACSVerse flow cytometer (BD Biosciences, San Jose, CA, USA) and Cytoflex (Beckman coulter) and analyzed using FlowJo 7.6 and FlowJo X software.

**Immunofluorescence (IF).** For IF, tumors were fixed in 4% paraformaldehyde, dehydrated in 30% sucrose, cut into slices by freezing-microtome, stained with the indicated fluorescently labeled antibodies and observed with a confocal microscope (LSM 880 with Airyscan, Carl Zeiss, Jena, Germany).

**Intravital microscopy.** For live-imaging of tumor tissue, dorsal-skin-fold window chamber EMT6 tumor-bearing mice were anaesthetized by intraperitoneally injecting pentobarbital, photothermally treated with 40 °C for 5 min or left untreated, and i.v. injected with 2 μg PE-Ly6G antibody and NPs@DiO (3 μg DiO). Then the mice were put on a coverslip with tumor above the ×20 objective lens (LSM 880 with Airyscan). After selecting a blood vessel in tumor, neutrophils in the blood vessel were photographed once every minute. These images were processed with Image J software.

**Enzyme-linked immunosorbent assay (ELISA).** For cytokine detection by ELISA, tumors were excised and homogenized in RIPA cell lysis buffer (Beyotime, Shanghai, China) with 1 mM phenylmethylsulfonyl fluoride. Moreover, the resulting homogenate was kept in ice to lyse cells for 45 min and centrifuged at $13,523 \times g$ for 20 min to get the total protein solution. The concentrations of CXCL1 and MIP-2 in tumors were assessed by a commercial ELISA Kit (Ray-Biotech, Norcross, GA, USA) according to the manufacturer's instructions.

To determine serum IL-6 and TNF-α levels after NPN injection, blood was collected at indicated time points and centrifuged to prepare serum. The concentrations of IL-6 and TNF-α in serum were measured by a commercial ELISA kit (Dakewe, Shenzhen, China) according to the manufacturer's instructions.

**Cell transfer and distribution in mice.** In all, $1 \times 10^7$ DiD-labeled neutrophils purified from bone marrow cells were i.v. injected into control or photothermally treated BALB/c mice ($n = 3–4$ for each group). At different time points, in vivo fluorescent images were observed with IVIS. Moreover, after 24 h, the major organs of mice, including heart, liver, spleen, lung, kidney, and tumors, were collected and fluorescent images of these organs were acquired with IVIS. For flow cytometric analysis, the tumors were harvested 10 h post-cell transfer and the percentage of DiD$^+$ neutrophils in CD45$^+$ cells was determined by flow cytometry.

**Distribution of NPs and NPNs in circulating and tumoral neutrophils.** NPs@DiO and NPNs@DiO were i.v. injected into photothermally treated (40 °C, 5 min) EMT6-bearing mice at an equivalent dose of 2.0 μg DiO ($n = 3–5$ for each group). Four hours later, neutrophils and other immune cells with NPs in blood and tumor were detected by flow cytometry.

To investigate the role of TLR4 and TLR2 in the recognition of NPNs by neutrophils, NPNs@DiD were i.v. injected into $Tlr4^{-/-}$ mice and TLR2-blocked mice by i.v. injecting anti-TLR2 antibody (20 μg, InvivoGen, San Diego, CA, USA). After 4 h, the percentage of neutrophils with NPNs in blood was determined by flow cytometry.

To explore whether NPN uptake would affect the migration of neutrophils to inflamed tumors, EMT6-bearing mice were treated with PTT and then i.v. injected with PBS, NPs, or NPNs. After 4 h, the ratio of neutrophils in the tumors (relative to tumor cells) was analyzed by flow cytometry.

**Distribution of NPs and NPNs in mice.** NPs@DiD and NPNs@DiD were i.v. injected into control and photothermally treated (40 °C, 5 min) EMT6-bearing mice at an equivalent dose of 1.8 μg DiD ($n = 3$ for each group). At different time points, in vivo fluorescent images were observed with IVIS. Moreover, after 48 h, the major organs of mice, including heart, liver, spleen, lung, kidney, and tumors, were collected and fluorescent images of these organs were acquired with IVIS.

**Formation of NETs by in vitro and in vivo neutrophils.** In all, $2 \times 10^5$ neutrophils were seeded on coverslips coated with poly-lysine in a 24-well plate and then incubated with 100 nM PMA for 5 h. After washing with PBS twice, the cells were fixed with 4% paraformaldehyde for 10 min at room temperature. Subsequently, the cells were stained with 5 μg mL$^{-1}$ PI for 30 min. After washing with PBS thrice, coverslips were mounted on glass microscope slides with a drop of anti-fade agent (Sigma-Aldrich) to reduce fluorescence photobleaching and then visualized with a confocal microscope (LSM 880 with Airyscan).

In vivo NETosis in control and PTT-treated EMT6 tumors 24 h after PTT was analyzed by IF. The tumor slides were stained with PE-Ly6G (5 μg mL$^{-1}$, Biolegend) and rabbit anti-Histone H3 citrulline (4 μg mL$^{-1}$, Abcam, Cambridge, UK) overnight after saturation with 1% BSA. Then the slides were further stained with FITC-anti rabbit antibody (6.67 μg mL$^{-1}$, Absin, Shanghai, China). After washing and DAPI staining, the slides were observed with a confocal microscopy (LSM 880 with Airyscan).

**Intercellular transport of NPs from neutrophils to tumor cells.** For in vitro experiment, EMT6 cells ($5 \times 10^4$ cells) were seeded on coverslips coated with poly-lysine in a 24-well plate. After 16 h of culture, EMT6 cells were incubated with 500 μL of DiD-labeled neutrophils encapsulated with NPs@DiO (NPs@DiO/DiD-neutrophils) ($1 \times 10^5$ cells) or 100 nM PMA-treated NPs@DiO/DiD-neutrophils. Five hours later, the cells were gently washed with PBS twice and fixed with 4% paraformaldehyde for 10 min at room temperature. Cells were then stained with 10 μg mL$^{-1}$ DAPI for 2 min to indicate cell nuclei. Coverslips were mounted on glass microscope slides with a drop of anti-fade agent to reduce fluorescence photobleaching and then visualized with a confocal microscope (LSM 880 with Airyscan).

For in vivo experiment, $1.7 \times 10^7$ DiD-labeled neutrophils encapsulated with NPs@DiO were intratumorally injected into untreated and photothermally treated EMT6 tumors ($n = 4$ per group). After 20 h, the tumors were excised and the percentages of DiO signals in transferred neutrophils and EMT6 tumor cells were analyzed by flow cytometry and IF. The experiments were also performed in mice with EMT6-EGFP tumors using transferred neutrophils encapsulating NPNs@DiD following the similar procedures.

**Tumor suppression study.** EMT6 tumor-bearing mice were randomly divided into ten groups ($n = 5$ per group) when the tumor volumes were ~60 mm$^3$. Mice were treated once with PBS, cisplatin, OMVs, NPs@Pt, NPNs@Pt, PTT, PTT plus cisplatin, PTT plus OMVs, PTT plus NPs@Pt, and PTT plus NPNs@Pt on day 0, respectively. The i.v. injection dose of cisplatin and OMVs was 2.0 mg per kg body weight and 0.2 mg per mouse, respectively, and the temperature of tumor site was kept at 40–41 °C for 5 min during PTT. Tumor volumes were monitored by measuring the perpendicular diameter using caliper every other day and tumor volume was calculated based on equation as follows: tumor volume = 0.5 × length × width$^2$. Mice were euthanized when the tumor size reached 2000 mm$^3$. We also measured the weight of mice every two days. At the end of experiment, the weight of tumor mass and the images of tumors were recorded. For the neutrophil depletion, EMT6-bearing mice were i.p. injected with 200 μg Ly6G antibody (α-Ly6G, 1A8 clone, BioXcell, West Lebanon, NH, USA) 1 day before receiving PTT plus NPNs@Pt. For the NETosis inhibition, EMT6-bearing mice were received two i.p. injections of BB-Cl-amidine; one injection at 30 min before PTT and the other one at 24 h after PTT (each injection 8 mg per kg).

For two treatments of PTT plus NPNs@Pt, EMT6 tumor-bearing mice were randomly divided into five groups ($n = 5–6$ per group) when the tumor volumes were ~80 mm$^3$. Mice were treated with PBS, NPs@Pt, NPNs@Pt, PTT plus NPs@Pt, and PTT plus NPNs@Pt, respectively, on day 0 and day 3. The i.v. injection dose of cisplatin and OMVs was 2.0 mg per kg body weight and 0.2 mg per mouse, respectively, and the temperature of tumor site was kept at 40–41 °C for 5 min during PTT. Tumor volumes were monitored by measuring the perpendicular diameter using caliper every other day and tumor volume was calculated based on equation as follows: tumor volume = 0.5 × length × width$^2$. We also measured the weight of mice every two days. At the end of experiment, the weight of tumor mass and the images of tumors were recorded. The therapeutic experiments were also performed in the CT26 tumor model.

**H&E staining and immunohistochemistry.** EMT6 tumor-bearing mice were treated once with PBS, cisplatin, OMVs, NPs@Pt, NPNs@Pt, PTT, PTT plus cisplatin, PTT plus OMVs, PTT plus NPs@Pt, and PTT plus NPNs@Pt on day 0, respectively, when the tumor volumes were ~60 mm$^3$. After 3 days, mice were euthanized and tumor tissues were removed. Tumors were fixed with 4% paraformaldehyde for 48 h, embedded in paraffin and sectioned into 5 μm section for H&E staining and immunohistochemistry. To detect the apoptosis and proliferation of tumor cells, TUNEL method (Roche, Basel, Switzerland) and Ki67 antibody (Cell signaling technology, Danvers MA, USA) staining were performed according to the manufacturer's instructions, respectively. In addition, at the end of two treatments of PTT plus NPNs@Pt, spleen, liver, and lung were harvested for H&E staining. To analyze the effect of tumor size on the PTT treatment, tumors of different sizes were excised at 4 h after PTT treatment. The tumors were sliced for H&E staining, Ki67 antibody (Cell signaling technology) staining, TUNEL staining (Roche), and Ly6G antibody (Biolegend) staining, respectively.

**Statistical analysis.** Data are expressed as the mean ± SD or mean ± SEM. An unpaired two-tailed Student t-test was used to analyze the statistically significant differences and data were considered statistically significant when the values of $P < 0.05$. *$P < 0.05$, **$P < 0.01$, ***$P < 0.001$, and ****$P < 0.0001$. n.s., not significant. N.D., not detected.

**Reporting summary**. Further information on research design is available in the Nature Research Reporting Summary linked to this article.

## Data availability

All data that support the findings of this study are available from the corresponding author on reasonable request. The raw data underlying Figs. 2–8 and Supplementary Figs. 1, 3, 7, 8, 10–12, 14, 17–19, 21, 24, 27, 29, 31–35 are provided as a Source Data file.

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

## Acknowledgements

This work was supported by the National Key R&D Program of China (2017YFA0205600), the National Natural Science Foundation of China (51961145109, 91942310, 51773191 and 81901683), the Fundamental Research Funds for the Central Universities (WK3520000009 and WK2070000126), the Open Project of Key Laboratory of Biomedical Engineering of Guangdong Province (KLBEMGD201701), and Anhui Provincial Natural Science Foundation (1908085QC100). This work was partially carried out at the USTC Center for Micro and Nanoscale Research and Fabrication.

## Author contributions

Y.Wang supervised the project. Y.Wang and M.L. conceptualized and designed the research. M.L., S.L., H.Z., X.T, Y.Wu, and W.J. performed the experiments. X.Y. provided the PBIBDF-BT polymer. Z.T. provided the *Tlr4*$^{-/-}$ mice. X.Z. provided help in data processing. M.L. and Y.Wang prepared the figures and wrote the manuscript. All authors read and approved the manuscript.

## Competing interests

The authors declare no competing interests.
