## [Peer Review File · Nature Communications]

Reviewers' comments:

Reviewer #1 (Remarks to the Author):

Li et al., is a well-presented study describing the use of micro-particles engineered to be coated with membranes from pathogens (nano-pathogenoid) as efficient means to target tumours with anti-tumour drugs. The study proposes a strategy involving circulating neutrophils as cellular vectors for the delivery of the nano-pathogenoid to the tumour. Overall the work is original, the experiments are well performed and it represents a proof of concept that could have important implication for the cancer field. The clarity in the presentation and the use of outstanding schematic drawing is certainly a strength of this manuscript, which also makes it very easy and pleasant to read.

Although some additional experiments (see below) are required to be able to link the effect to the neutrophils' activity to the observed effects, in this reviewer opinion, the study should be considered for publication in Nature Communication, provided the authors will satisfy the comments listed below.

Main Comments

1. The main experimental setting missing in the study is the one directly linking the effect of combining MPNs@Pt injections and PTT exposure with neutrophils presence. The authors have done a great deal of work to show that neutrophils accumulate in tumour as consequence of PTT exposure. They have also shown that neutrophils are able to pick up the nanoparticles injected in the circulation. However, to evaluate the actual functional impact of the neutrophils mediated delivery of nanoparticles, the experimental setting should include at least one of the following:

- Option 1- neutrophils depletion (using the Ly6G antibody). Even if the experiments are performed using Balb/c mice, which have a rather high level of circulating neutrophils, the use of the Ly6G blocking antibody (1A8 clone) should be sufficiently efficient to deplete neutrophils for at least a 7 day period. This period would cover the steps of IVs and PTT treatments, a time window when blocking circulating neutrophils from picking up the MPNs@Pt particles should reduce the efficacy of the intervention.
- Option 2- the authors could use the CXCR2 ko mice (which are also available in Balb/c background) where neutrophils do not respond to CXCR2-CXCL1 signals driving their recruitment. If their accumulation to the tumour is CXCR2 dependent, no neutrophils increased should be detected upon PTT. This setting would also allow to mechanistically link neutrophils accumulation with the CXCL1 shown to increase in PTT treated tumours (Figure 2I).

If neutrophils are functionally crucial for the enhancement of the tumour killing activity of the MPNs@Pt, the results of either of the two settings should show a significant difference in the outcome when treating tumour with PTT in absence of neutrophils recruitment.

2. With the current data, the increase of CXCL1 and MIP-2 in the tumour upon PTT treatment it is not sufficient to support the model described on page 8, line 152-154 “[...] neutrophils migrated to PTT-treated tumours along chemokine gradients of CXCL1 and MIP-2 [...]”. No direct experiments interfering with those chemokines are provided, therefore, at the moment, this can only be a plausible hypothesis.

3. On the same line, There are not direct evidences that neutrophils deliver the particles to the tumour cells via NETosis. The images in Figure 6G do not show NETosis in neutrophils in vivo and, at this magnification, it is also not possible to estimate the co-localization of the DiO in neutrophils or cancer cells. Suitable higher magnifications need to be provided to show the DiO co-localization. Moreover, a dedicated staining for NETs should also be shown such as Cit-histone H3 to determine if the phenomenon is really triggered by PTT treatment. In which case an hypothesis can be made that NETosis could be involved in Neutrophils delivery. However, to formally show that this is the case, NETosis inhibitor (such as PAD4 inhibitor) should be used to show a reduction of treatment activity.

4. Concerning Figure 2i the authors comments on the fact that PTT treated EMT6 tumour with different size have difference abundance of neutrophils infiltration, and hypothesised an insufficient tissue penetration of NIR or impaired blood vessel structure in bigger tumours. However, to determine the effect of PTT treated on neutrophils recruitment, the histology of the tumours in Figure 2i should be provided. The study only shows histology in Figure 7, but it would be very important early on to show the consequences of PTT treatment on cancer cells, which is a key approach in this work. The overall H&E staining, Ki67 and a marker of cell death, along side with neutrophils staining (such S100A9) should be presented to define the consequence of PTT treatment in tumours of different sizes.

5. From the setting in figure 7 and figure 8 two things need to be clarified:

- Is there a difference between the IV injection done before the PTT treatment versus the one done at the same time? In the text and legend, it looks like they are the various formulations in both cases, but in the schematics, it looks like different things are being injected. Please clarify.
- How big are tumour when they are treated with PTT, after 8 days of inoculation?

Other Comments

6. To validate the potential clinically relevant outcome of Figure 8, the authors should repeat the experiment also with the CT26 colon cell line. This will also provide important information in case the efficiency of the treatment would be different. Given the reduced presence of Neutrophils in this two models (highlighted in Figure 2i in EMT6 vs CT26 tumours of similar size) one would be able to consider that the potential impact of the steady state neutrophils presence in tumours can predict the therapeutic efficacy of this approach.

7. Histology of spleen liver and lung for the end point (20 day) of the treated mice in the experiment shown in Figure 8 should be provided to estimate the potential damage induced by the treatment. Considering that neutrophils presence, even if it does not increase as consequence of PTT treatment, it is quite high in those tissue (as supported by the data in Figure 3d,e), the expectation would be

that if neutrophils collect the particle in the circulation could also accumulate them in high amount to those tissue.

Reviewer #2 (Remarks to the Author):

The manuscript by Li et al describes a new drug delivery strategy that combines blood neutrophil targeting using bacterial outer membrane vesicles (OMVs) with photothermal therapy, which promotes neutrophil migration into tumor sites. Overall, the concept is innovative and the experiments are well controlled and properly conducted. While the idea of cell hitchhiking is not new, it is still a developing field with room for new approaches and this strategy for targeting neutrophils is innovative and appears to work well. Therefore, there are broad potential applications of this approach for improving delivery of diverse drugs. Therefore, based on the innovative idea, the rigor of most of the studies, and the significance of the work to the drug delivery and cancer nanotechnology communities, the work appears suitable for publication in Nature Communications provided the following comments are addressed:

1. In Figure 5, a comparison between NP and NPN (for example loaded with a dye) should be made in terms of total tumor uptake. As is, the data shows that NPNs target neutrophils and that more neutrophils home to tumors after PTT, but it is not clearly shown to what extent an NP or its cargo will accumulate in a tumor when the NPN approach is used versus just a simple NP formulation (both with and without PTT). While the data points to this conclusion, the study necessary to properly conclude this does not appear to have been performed. This is an important study to do as it links improved delivery efficiency to the design.
2. It is important to determine what the receptor on neutrophils is for the NPN. LPS is mentioned but this was not shown; for example, blocking TLR4 could be done to show this. Or other receptors. At minimum, more discussion about what PAMPs may be mediating this interaction is important. Ultimately, knowledge of the receptors being utilized here may motivate future efforts to maximize this effect.
3. In Figure 5h, only tumor-associated neutrophils are analyzed. Many cells in tumors can take up nanoparticles, including macrophages. What percentage of the major immune cell types is NP+ within the tumor microenvironment? This is important to characterize.

4. The idea that netosis results in NP/drug release is compelling. The data in Figure 6 would be more compelling if flow cytometry was also performed to show i) reduced NP in neutrophils under PTT conditions, ii) NP uptake by other cell populations. Furthermore, it is not entirely clear that it is tumor cells that are taking up the NPs after netosis - these could be other immune cells like macrophages just as well. It is necessary to more conclusively prove that tumor cells take up the NPs; for example, by using a GFP+ tumor line and co-localizing with NP/dye.

5. Considering that LPS is injected IV, levels of serum cytokines, such as IL-6, TNF- α , etc should be measured. The toxicity appears minor but this would nonetheless be important to describe.

Reviewer #3 (Remarks to the Author):

Recommendation: Publish in Nature Communications after minor revisions noted.

In this manuscript, Li et al. established a pathogen-mimicking nano-pathogenoid (NPN) system to hitchhike circulating neutrophils in vivo for tumor-targeted drug delivery. They showed that the circulating neutrophils were able to accumulate and penetrate in tumors with photothermal therapy (PTT) enhanced inflammation in response to increased level of chemokines such as CXCL1. They also demonstrated that the neutrophils in circulation could internalize NPN which were transported to the tumors and released in situ partially during NETosis of the neutrophils. When cisplatin was loaded within the NPN and used in combination with PTT, increased tumor growth inhibition was observed after a single dosage, and tumor regression was achieved after repeated treatments. This bio-inspired strategy provided a novel strategy for neutrophil targeting in vivo and exhibited great potential and effectiveness in cancer therapy, which could be also possibly used in treating infectious and autoimmune diseases. In general, the present work merit publication in Nature Communications after addressing the following minor issues.

1. Successful coating of OMVs onto nanoparticles is a prerequisite for neutrophils to capture and deliver NPs. There was no related data demonstrating the structure of NPN in the manuscript. Please provide.
2. Extrinsic particles internalized by neutrophils usually undergo destruction and deactivation. Please verify the cytotoxicity of naïve NPN and NPV released by neutrophils.
3. Please verify whether the migration of neutrophils was affected by captured NPNs, as this capability is essential for tumor-targeted drug delivery.

4. PTT at a higher temperature such as 42 °C would exert a stronger cell eradication effect and be more efficient recruitment of neutrophils (Fig. 2j). What is the rationale of maintaining intratumoral temperature at ~40 °C during in vivo study? Please discuss.

5. In Fig. 4, intravital imaging was performed to monitor the behavior of neutrophils in the tumors in real-time. Although very convincing, the results reflected the behavior of neutrophils in tumor tissue with functional perfusion. Please discuss how did the neutrophil migration and NPN diffusion affect tumor tissues without effective perfusion.

6. In Fig. 8j and supplementary Fig 19, the result showed that OMV injection resulted in a transient weight loss of mice, indicating some toxicity of OMV, how to solve the problem? Please comment.

Reviewer #1 (Remarks to the Author):

Comments:

Li et al., is a well-presented study describing the use of micro-particles engineered to be coated with membranes from pathogens (nano-pathogenoid) as efficient means to target tumours with anti-tumour drugs. The study proposes a strategy involving circulating neutrophils as cellular vectors for the delivery of the nano-pathogenoid to the tumour. Overall the work is original, the experiments are well performed and it represents a proof of concept that could have important implication for the cancer field. The clarity in the presentation and the use of outstanding schematic drawing is certainly a strength of this manuscript, which also makes it very easy and pleasant to read. Although some additional experiments (see below) are required to be able to link the effect to the neutrophils' activity to the observed effects, in this reviewer opinion, the study should be considered for publication in Nature Communication, provided the authors will satisfy the comments listed below.

Author reply: We thank this reviewer for his/her very positive assessment on this work and we have carefully addressed the issues raised by the reviewer.

Comments 1: The main experimental setting missing in the study is the one directly linking the effect of combining NPNs@Pt injections and PTT exposure with neutrophils presence. The authors have done a great deal of work to show that neutrophils accumulate in tumour as consequence of PTT exposure. They have also shown that neutrophils are able to pick up the nanoparticles injected in the circulation. However, to evaluate the actual functional impact of the neutrophils mediated delivery of nanoparticles, the experimental setting should include at least one of the following:

- Option 1- neutrophils depletion (using the Ly6G antibody). Even if the experiments are performed using Balb/c mice, which have a rather high level of circulating neutrophils, the use of the Ly6G blocking antibody (1A8 clone) should be sufficiently efficient to deplete neutrophils for at least a 7 day period. This period would cover the steps of IVs and PTT treatments, a time window when blocking circulating neutrophils from picking up the NPNs@Pt particles should reduce the efficacy of the intervention.
- Option 2- the authors could use the CXCR2 ko mice (which are also available in Balb/c background) where neutrophils do not respond to CXCR2-CXCL1 signals driving their recruitment. If their accumulation to the tumour is CXCR2 dependent, no neutrophils increased should be detected upon PTT. This setting would also allow to mechanistically link neutrophils accumulation with the CXCL1 shown to increase in PTT treated tumours (Figure 2l).

If neutrophils are functionally crucial for the enhancement of the tumour killing activity of the NPNs@Pt, the results of either of the two settings should show a significant difference in the outcome when treating tumour with PTT in absence of neutrophils recruitment.

Author reply: We thank the reviewer for his/her valuable suggestions. As suggested, we evaluated the therapeutic efficacies of PTT + NPNs@Pt after depleting neutrophils with Ly6G antibody (α -Ly6G, 1A8 clone). As shown in **Supplementary Fig. 34a**, a single *i.p.* injection of α -Ly6G (200 μ g) was able to deplete neutrophils for 7 days. The absence of neutrophils compromised the therapeutic efficacy (tumor inhibition efficiency: 96.0% for PTT + NPNs@Pt versus 58.0% for α -Ly6G + PTT + NPNs@Pt) (**Supplementary Fig. 34b, c**). And neutrophil depletion also reduced the survival rate of mice (**Supplementary Fig. 34d**), proving the necessity and vital role of

neutrophils as cell carriers in the tumor combating effect of NPNs@Pt plus PTT. In addition, no body weight reduction was observed after neutrophil depletion (**Supplementary Fig. 34e**), suggesting that such differences were not caused by the toxicity of α -Ly6G. These data was included in the supplementary information as **Supplementary Fig. 34** in the revision and relevant discussion was added in the revised manuscript (page 17, line 16-20).

Supplementary Fig. 34 Neutrophil depletion impaired the therapeutic efficacy of PTT plus NPNs@Pt. **a** The percentage of neutrophils in blood at indicated time points after receiving an *i.p.* injection of 200 μ g α -Ly6G into mice. $n = 3$. **b** Workflow for determining the role of neutrophils in the therapeutic efficacy of PTT plus NPNs@Pt on EMT6-bearing mice. On day -1 (defined as the 8th day after tumor inoculation), the mice were received an *i.p.* injection of α -Ly6G followed by an *i.v.* injection of NPNs@PBT. On day 0, the mice were received PTT treatment (40 °C, 5 min) on the tumors followed by *i.v.* injection of NPNs@Pt (2 mg cisplatin per kg body weight). Tumor volumes were measured every two days from day 0 to day 18. (**c-e**) Tumor growth curves (**c**), percent of survival (**d**), and mouse body weight (**e**) during the experiment. $n = 5$ per group. Data are shown as mean \pm SEM and analyzed by unpaired Student's *t*-test. *** $P < 0.001$. Source data are provided as a Source Data file.

2. With the current data, the increase of CXCL1 and MIP-2 in the tumour upon PTT treatment it is not sufficient to support the model described on page 8, line 152-154 “[...] neutrophils migrated to PTT-treated tumours along chemokine gradients of CXCL1 and MIP-2 [...]”. No direct experiments interfering with those chemokines are provided, therefore, at the moment, this can only be a plausible hypothesis.

Author reply: We thank the reviewer for his/her valuable suggestions. It has been widely reported that CXCL1 and MIP-2 are two major chemokines responsible for recruiting neutrophils by binding to the chemokine receptor CXCR2 (*Blood*, **121** 4930-4937 (2013); *Cancer Cell*, **29** 832-845 (2016); *Science*, **330** 362-366 (2010)). We used SB225002, a small molecule inhibitor of the receptor CXCR2 (CXCR2i) to block its interaction with CXCL1 and MIP-2 and to verify the involvement of these chemokines in the PTT-induced neutrophil recruitment (*Gut*, **67** 1112-1123

(2018); *Cancer Res*, **78** 5586-5599 (2018)). Inhibition of CXCR2 by SB225002 significantly decreased the recruitment of neutrophils to PTT-treated tumors, demonstrating the involvement of CXCL1 and MIP-2 in the neutrophil chemotaxis (**Supplementary Fig. 8**). The data was included in the supplementary information as **Supplementary Fig. 8** in the revision and relevant discussion was added in the revised manuscript (page 8, line 10-13).

Supplementary Fig. 8. Inhibition of CXCR2 decreased neutrophil recruitment to PTT-treated tumors. EMT6-bearing mice were treated with PTT on tumors 30 min post injection of CXCR2 inhibitor SB225002 (CXCR2i). Mice pre-injected with 0.33% tween-80 and 10% DMSO in PBS were used as the control. After 4 h, tumors were excised and analyzed by flow cytometry. **a** Gating strategy to analyze populations of neutrophils and tumor cells in the tumors. **b** The ratio of neutrophils to tumor cells in the tumors after the indicated treatments. $n = 8$ per group. Data are shown as mean \pm SEM and analyzed by unpaired Student's t -test. $*P < 0.05$. Source data are provided as a Source Data file.

3. On the same line, there are not direct evidences that neutrophils deliver the particles to the tumour cells via NETosis. The images in Figure 6G do not show NETosis in neutrophils in vivo and, at this magnification, it is also not possible to estimate the co-localization of the DiO in neutrophils or cancer cells. Suitable higher magnifications need to be provided to show the DiO co-localization. Moreover, a dedicated staining for NETs should also be shown such as Cit-histone H3 to determine if the phenomenon is really trigger by PTT treatment. In which case an hypothesis can be made that NETosis could be involved in Neutrophils delivery. However, to formally show that this is the case, NETosis inhibitor (such as PAD4 inhibitor) should be use to show a reduction of treatment activity. **Author reply:** We thank the reviewer for his/her valuable suggestions. As suggested by the reviewer, we have magnified Figure 6g and also added zoom-in images in the revision. The current revised **Figure 6g** clearly showed that DiO-labelled NPs were mostly contained within DiD-labelled neutrophils in control tumors. By contrast, in PTT-treated tumors, NPs were released and uptaken by tumor cells.

Figure 6g. DiO signals in transferred neutrophils and EMT6 tumor cells were analyzed by IF. Yellow arrows indicated NPs@DiO co-localized with DiD-labelled neutrophils, while green arrows indicated NPs@DiO co-localized with EMT6 tumor cells. Scale bars, 10 μm for the magnified images, and 50 μm for all other images.

In addition, we evaluated the micro-distribution of NPs in tumors derived from EGFP expressing EMT6 cells (EMT6-EGFP) to further verify the transfer of NPs from neutrophils to tumor cells after PTT. In PTT-treated tumors, more NPNs@DiD were observed within EMT6-EGFP tumor cells (**Supplementary Fig. 30**), further proving that tumor cells could internalize the NPNs released by neutrophils. The data was included in the supplementary information as **Supplementary Fig. 30** in the revision and the relevant discussion was added in the revised manuscript (page 15, line 13-17).

Supplementary Fig. 30 EGFP expressing EMT6 tumor cells (EMT6-EGFP) internalized NPNs@DiD released by neutrophils in PTT-treated tumors. EMT6-EGFP bearing mice were left untreated or treated with PTT (40 °C for 5 min). After 30 min, neutrophils encapsulated with DiD-labelled NPNs (NPNs@DiD) were intratumorally injected into the mice. After another 9 h, DiD fluorescent signals were analyzed by immunofluorescence. Scale bars, 20 μ m for the amplification panels, and 100 μ m for all other panels.

Moreover, we sought to verify the formation of NETs during PTT treatment by visualizing the co-localization of Ly6G and citrullinated histone H3 (Cit His H3), a widely accepted marker of NETs (*Nat. Commun.*, **7** 10973-10985 (2016)) (**Supplementary Fig. 28**). PTT resulted in more infiltration of neutrophils accompanied by higher amount of Cit His H3 in the tumors, which confirmed the formation of NETs. These data was included in the supplementary information as **Supplementary Fig. 28** in the revision and relevant discussion was added in the revised manuscript (page 15, line 3-5).

Supplementary Fig. 28 *In vivo* NETosis in PTT-treated tumor. EMT6-bearing mice were left untreated or treated with PTT (40 °C for 5 min). After 24 h, the tumors were excised and were stained with Ly6G antibody and antibody of citrullinated histone H3 (Cit His H3) to analyze *in vivo* NETosis after PTT treatment. Scale bars, 20 μ m.

Next, to assess whether NETosis was involved in the combined therapy of PTT with NPNs@Pt, we adopted PAD inhibitor BB-Cl-amidine to inhibit NETosis (*Nat. Med.*, **23** 279-287 (2017); *Nat. Commun.*, **9** 4783-4797 (2018)) (**Supplementary Fig. 35a**). The inhibition of NETosis substantially accelerated the tumor growth compared to mice received treatment of PTT + NPNs@Pt but without PAD inhibitor injection (**Supplementary Fig. 35b**). In line with this, NETosis inhibition also reduced the survival rate of mice, indicating the importance of NETosis for the excellent treatment efficacy of PTT with NPNs@Pt (**Supplementary Fig. 35c**). In addition, only a moderate and transient body weight reduction was observed after BB-Cl-amidine injection (**Supplementary Fig. 35d**), suggesting that such differences in the therapeutic outcomes were not caused by the toxicity of BB-Cl-amidine. These data was included in the supplementary information as **Supplementary Fig. 35** in the revision and relevant discussion was added in the revised manuscript (page 17, line 20-22, and page 18, line 1-3).

Supplementary Fig. 35 Inhibiting NETosis with BB-Cl-amidine impaired the therapeutic efficacy of PTT with NPNs@Pt. **a** Workflow for determining the role of NETosis in the therapeutic efficacy of PTT plus NPNs@Pt on EMT6-bearing mice. On day -1 (defined as the 8th day after tumor inoculation), the mice were injected with NPNs@PBT. On day 0, mice were *i.p.* injected with 160 μg BB-Cl-amidine to inhibit NETosis. After 30 min, the mice were received PTT treatment (40 $^{\circ}\text{C}$, 5 min) on the tumors followed by *i.v.* injection of NPNs@Pt (2 mg cisplatin per kg body weight). On day 1, the mice were received an additional injection of 160 μg BB-Cl-amidine. Tumor volumes were measured every two days from day 0 to day 18. (**b-d**) Tumor growth curves (**b**), percent of survival (**c**), and mouse body weight (**d**) during the experiment. $n = 5$ per group. Data are shown as mean \pm SEM and analyzed by unpaired Student's *t*-test. *** $P < 0.001$. Source data are provided as a Source Data file.

4. Concerning Figure 2i the authors comments on the fact that PTT treated EMT6 tumour with different size have difference abundance of neutrophils infiltration, and hypothesized an insufficient tissue penetration of NIR or impaired blood vessel structure in bigger tumours. However, to determine the effect of PTT treated on neutrophils recruitment, the histology of the tumours in Figure 2i should be provided. The study only shows histology in Figure 7, but it would be very important early on to show the consequences of PTT treatment on cancer cells, which is a key approach in this work. The overall H&E staining, Ki67 and a marker of cell death, along site with neutrophils staining (such S100A9) should be presented to define the consequence of PTT treatment in tumours of different sizes.

Author reply: As suggested by the reviewer, we performed additional experiments to determine the effect of PTT on tumor killing and neutrophil recruitment in tumors with different sizes (~ 60 and 240 mm^3). Tumors with sizes of ~ 60 and 240 mm^3 were treated with PTT (40 $^{\circ}\text{C}$ for 5 min) or left untreated. After 4 h, tumors were obtained and dissected to observe the cellular apoptosis and proliferation as well as neutrophil infiltration at different depths inside the tumor tissues (**Supplementary Fig. 5a**). Due to the insufficient tissue penetration of the NIR laser, the tumor killing efficacy of PTT decreased as tumor volume increased, as evidenced by decreased apoptotic/necrotic area (H&E and terminal deoxynucleotidyl transferase-mediated dUTP nick end-labeling (TUNEL) staining, **Supplementary Fig. 5b and 5c**) and increased Ki67 positive cells (**Supplementary Fig. 5d**). Consistent with this, the infiltration of neutrophils decreased in

tumors with larger sizes (**Supplementary Fig. 5e**). These data was included in the supplementary information as **Supplementary Fig. 5** in the revision and relevant discussion was added in the revised manuscript (page 7, line 11-15).

Supplementary Fig. 5 PTT induced less cellular apoptosis and neutrophil recruitment in the deeper regions of larger tumors. EMT6 tumors with sizes of ~60 or 240 mm³ were treated with PTT (40 °C for 5 min) or left untreated as a control. After 4 h, the tumors were obtained and dissected to observe the cellular apoptosis and proliferation as well as neutrophil infiltration at different depths inside the tumor tissues. **a** Schematic showing analyses of PTT effects in tumor regions at different depths under the irradiated skin. **(b-e)** Representative images of **(b)** H&E, **(c)** terminal deoxynucleotidyl transferase-mediated dUTP nick end-labeling (TUNEL), **(d)** Ki67, and **(e)** Ly6G staining of tumor slices obtained at different depths inside the tumors. Scale bar, 100 μm.

5. From the setting in figure 7 and figure 8 two things need to be clarified:

- Is there a difference between the IV injection done before the PTT treatment versus the one done at the same time? In the text and legend, it looks like they are the various formulations in both cases, but in the schematics, it looks like different things are being injected. Please clarify.
- How big are tumour when they are treated with PTT, after 8 days of inoculation?

Author reply: We thank for the reviewer for pointing this out. We actually injected different formulations before and after the PTT. In specific, before the PTT treatment, NPs@PBT as the PTT transducer were injected. While after the PTT treatment, different formulations of cisplatin and OMVs were injected for the tumor treatment. We have added the information of injected formulations in the revised Figure 7a and 8a. We also revised the figure captions accordingly to make it clearer in the revised manuscript (page 53, 54).

Figure 7a.

Figure 8a.

Regarding the tumor size, the tumors are ~60 and 80 mm³ for the experiments in Figure 7 and Figure 8, respectively. We also added this information in the revised manuscript on page 16, line 2, and page 18, line 8, respectively.

6. To validate the potential clinically relevant outcome of Figure 8, the authors should repeat the experiment also with the CT26 colon cell line. This will also provide important information in case the efficiency of the treatment would be different. Given the reduced presence of Neutrophils in this two models (highlighted in Figure 2i in EMT6 vs CT26 tumours of similar size) one would be able to consider that the potential impact of the steady state neutrophils presence in tumours can predict the therapeutic efficacy of this approach.

Author reply: We thank the reviewer for his/her valuable suggestions. As suggested, we performed tumor treatment therapy in CT26 tumor model similar to that of Figure 8. On days 0 and 3 (defined as the 10th and 13th day after tumor inoculation), the mice received two indicated treatments. Tumor volumes were measured every two days from day 0 to day 12 (**Supplementary Fig. 33a**). Tumor volume results showed that treatment with PTT plus NPNs@Pt demonstrated significant tumor growth inhibition, compared to the groups of PBS, NPs@Pt, and NPNs@Pt (**Supplementary Fig. 33b**). However, there was no significant difference on tumor volume between groups of PTT plus NPs@Pt and PTT plus NPNs@Pt, which might be caused by the limited advantage of tumor accumulation of NPNs@Pt due to the low percentage of neutrophils in CT26 tumor. Tumor images (**Supplementary Fig. 33c**) and tumor weight (**Supplementary Fig. 33d**) showed similar results of tumor volume. Administration of OMVs caused a transient drop in the mouse weight, which, however, was recovered within 5 days (**Supplementary Fig. 33e**). These data indicate that the combined treatment of PTT with NPNs@Pt is more suitable for tumors with abundant neutrophils. These data was included in the supplementary information as **Supplementary Fig. 33** in the revision and relevant discussion was added in the revised manuscript (page 17, line 12-16).

Supplementary Fig. 33 Combined therapy of PTT with NPNs@Pt in CT26 tumor model. **a** Workflow for treatment of CT26-bearing mice using PTT-based neutrophil-mediated NPNs@Pt delivery system. On days 0 and 3 (defined as the 10th and 13th day after tumor inoculation), the mice received PTT treatment on the tumors followed by an *i.v.* injection of PBS or different formulations of cisplatin. Tumor volumes were measured every two days from day 0 to day 12. $n = 5$ per group. **b** Tumor growth curves during the treatments. **(c)** Photos and **(d)** weight of the tumors collected on day 13. **e** Change of mouse weight during the treatments. Data are shown as mean \pm SEM and analyzed by unpaired Student's *t*-test. * $P < 0.05$, ** $P < 0.01$. *n.s.*, not significant. Source data are provided as a Source Data file.

7. Histology of spleen, liver and lung for the end point (20 day) of the treated mice in the experiment shown in Figure 8 should be provided to estimate the potential damage induced by the treatment. Considering that neutrophils presence, even if it does not increase as consequence of PTT treatment, it is quite high in those tissue (as supported by the data in Figure 3d, e), the expectation would be

that if neutrophils collect the particle in the circulation could also accumulate them in high amount to those tissue.

Author reply: We appreciate this reviewer’s valuable suggestions. It should be noted that Figure 3d and e reflected the biodistribution of adoptively transferred neutrophils, instead of circulating neutrophils hitchhiked by NPNs. The adoptively transferred neutrophils are more likely to be phagocytosed by macrophages rich in mononuclear phagocyte system (MPS) in the liver and spleen due to the reduced cell viability after *in vitro* purification and processing. By contrast, for all the other Figures including Fig. 8, circulatory neutrophils were used without *in vitro* isolation and processing. In these cases, the accumulation of NPNs and neutrophils in the tumors was much higher than that in other organs, due to low MPS clearance and high targeting efficacy to the inflamed tumors (Figure 5l, m). Moreover, for the *in vivo* therapy experiments in Fig. 8, cisplatin was administrated at low dose and very low frequency (2 mg/mg per injection and 2 injections within 3 weeks). Hence, we would not expect toxicity in those normal tissues.

To further prove this and also as suggested by the reviewer, we performed histological examination of spleen, liver and lung obtained at the end of therapy (day 20). The results showed that all the treatments were of high biosafety without causing obvious damage to major tissues (**Supplementary Fig. 36**). This data was included in the supplementary information as **Supplementary Fig. 36** in the revision and relevant discussion was added in the revised manuscript (page 19, line 5-7).

Supplementary Fig. 36 Representative images of H&E staining of liver, spleen, and lung obtained at the end point of experiment. Scale bars, 100 μm .

Reviewer #2 (Remarks to the Author):

The manuscript by Li et al describes a new drug delivery strategy that combines blood neutrophil targeting using bacterial outer membrane vesicles (OMVs) with photothermal therapy, which promotes neutrophil migration into tumor sites. Overall, the concept is innovative and the experiments are well controlled and properly conducted. While the idea of cell hitchhiking is not new, it is still a developing field with room for new approaches and this strategy for targeting neutrophils is innovative and appears to work well. Therefore, there are broad potential applications of this approach for improving delivery of diverse drugs. Therefore, based on the innovative idea, the rigor of most of the studies, and the significance of the work to the drug delivery and cancer nanotechnology communities, the work appears suitable for publication in Nature Communications provided the following comments are addressed:

Author reply: We thank this reviewer for his/her positive assessment on this work and we have carefully addressed the issues raised by the reviewer.

1. In Figure 5, a comparison between NP and NPN (for example loaded with a dye) should be made in terms of total tumor uptake. As is, the data shows that NPNs target neutrophils and that more neutrophils home to tumors after PTT, but it is not clearly shown to what extent an NP or its cargo will accumulate in a tumor when the NPN approach is used versus just a simple NP formulation (both with and without PTT). While the data points to this conclusion, the study necessary to properly conclude this does not appear to have been performed. This is an important study to do as it links improved delivery efficiency to the design.

Author reply: We thank the reviewer for his/her valuable suggestions. We assessed the accumulation of DiD-labelled NPs and NPNs in tumors with or without PTT using IVIS. The tumor targeting ability of the neutrophil-hitchhiking NPNs after PTT was higher than that of EPR effect-based PEG-*b*-PLGA NPs. We also observed that PTT increased the accumulation of NPs, possibly caused by dilated tumor vasculature and increased blood perfusion after PTT. This data was included in the supplementary information as **Supplementary Fig. 24** in the revision and relevant discussion was added in the revised manuscript (page 14, line 4-6).

Supplementary Fig. 24 PTT enhanced tumor accumulation of NPNs compared with PEG-*b*-PLGA NPs. DiD labelled NPs and NPNs were *i.v.* injected into PTT-treated or untreated EMT6-bearing mice. **a** At the indicated time points, *in vivo* DiD fluorescent signals were observed with IVIS. **b** Quantitative ROI analysis of DiD fluorescent signals in tumor areas. $n = 3$ per group. Data are shown as mean \pm SEM and analyzed by unpaired Student's *t*-test. * $P < 0.05$, ** $P < 0.01$, *** $P < 0.001$, and **** $P < 0.0001$. *n.s.*, not significant. Source data are provided as a Source Data file.

2. It is important to determine what the receptor on neutrophils is for the NPN. LPS is mentioned but this was not shown; for example, blocking TLR4 could be done to show this. Or other receptors. At minimum, more discussion about what PAMPs may be mediating this interaction is important. Ultimately, knowledge of the receptors being utilized here may motivate future efforts to maximize this effect.

Author reply: We thank the reviewer for his/her valuable suggestions. We agree with the reviewer that it is important to determine what receptors on neutrophils mediated their specific interaction with NPNs. As suggested by this reviewer, we have performed additional *in vivo* experiments to investigate the roles of toll-like receptors TLR4 and TLR2 in recognizing NPNs. TLR2 and TLR4 are known to mediate response of neutrophils to bacteria by recognizing the lipoprotein and LPS on bacteria, respectively (*Nat Rev Immunol* **10**, 131-144 (2010)). In TLR4 knockout mice (TLR4^{-/-}), the uptake of NPNs by circulating neutrophils was significantly inhibited (Supplementary Fig. 17a, b). In addition, blockade of TLR2 using anti-TLR2 antibody decreased the neutrophil targeting efficiency by 25% (Supplementary Fig. 17c, d). The results suggested that both receptors, especially TLR4, played a predominant role in the recognition of NPNs by neutrophils. This data was included

in the supplementary information as **Supplementary Fig. 17** in the revision and relevant discussion was added in the revised manuscript (page 12, line 16-22).

Supplementary Fig. 17 Uptake of NPNs@DiD by neutrophils in TLR4 knockout mice (TLR4^{-/-}) and mice pre-treated with anti-TLR2 antibody. **a** Flow cytometric analyses of the percentage of neutrophils that contained NPNs@DiD (defined as Ly6G⁺DiD⁺ cells). Wild type (WT) mice or TLR4^{-/-} mice were injected with NPNs@DiD and subjected to flow cytometric analyses at 4 h post injection. **b** The inhibition rate of NPNs uptake by neutrophils in TLR4^{-/-} mice relative to WT mice. *n* = 4 per group. **c** Flow cytometric analyses of the percentage of neutrophils that contained NPNs@DiD (defined as Ly6G⁺DiD⁺ cells) in mice pre-treated with anti-TLR2 antibody. The mice were injected with NPNs@DiD and subjected to flow cytometric analyses at 4 h post injection. **d** The inhibition rate of NPNs uptake by neutrophils in anti-TLR2 treated mice relative to the untreated mice. *n* = 4 per group. Data are shown as mean ± SEM. Source data are provided as a Source Data file.

3. In Figure 5h, only tumor-associated neutrophils are analyzed. Many cells in tumors can take up nanoparticles, including macrophages. What percentage of the major immune cell types is NP⁺ within the tumor microenvironment? This is important to characterize.

Author reply: We thank the reviewer for his/her valuable suggestions. As suggested by this reviewer, we analyzed the internalization of NPs and NPNs by macrophages and other cells in PTT-treated tumors. As shown in **Supplementary Fig. 21**, besides neutrophils, most nanoparticles were predominately uptaken by macrophages rather than other immune cells. This data was included in the supplementary information as **Supplementary Fig. 21** in the revision and relevant discussion was added in the revised manuscript (page 13, line 14-16).

Supplementary Fig. 21 Uptake of NPs@DiO and NPNs@DiO by macrophages and other non-neutrophil immune cells in PTT-treated tumors. DiO-labelled NPs and NPNs were *i.v.* injected into PTT-treated EMT6-bearing mice. At 4 h post injection the percentage of macrophages and other non-neutrophil immune cells encapsulating NPs and NPNs was analyzed by flow cytometry. $n = 3$ for NPs group and $n = 5$ for NPNs group. Data are shown as mean \pm SEM. Source data are provided as a Source Data file.

4. The idea that netosis results in NP/drug release is compelling. The data in Figure 6 would be more compelling if flow cytometry was also performed to show i) reduced NP in neutrophils under PTT conditions, ii) NP uptake by other cell populations. Furthermore, it is not entirely clear that it is tumor cells that are taking up the NPs after netosis - these could be other immune cells like macrophages just as well. It is necessary to more conclusively prove that tumor cells take up the NPs; for example, by using a GFP+ tumor line and co-localizing with NP/dye.

Author reply: We thank the reviewer for his/her valuable suggestions. Flow cytometric analysis showed that the percentage of transferred neutrophils containing NPs@DiO decreased by 77.6% compared to the control group, proving that PTT induced the efficient release of cargos from neutrophils. Moreover, more NPs@DiO were taken up by EMT6 tumor cells after PTT. The data was included in the revised manuscript as **Figure 6f** and **Supplementary Fig. 29**. The relevant discussion was added in the revised manuscript (page 15, line 9-12 and 13-15).

Figure 6f. The percentage of transferred neutrophils containing NPs (defined as NP⁺DiD⁺) was detected by flow cytometry. *n* = 4 per group. Data are shown as mean ± SEM and analyzed by unpaired Student's *t*-test. ***P* < 0.01. Source data are provided as a Source Data file.

Supplementary Fig. 29 Tumor cells internalized NPs that were released from neutrophils after the PTT treatment. EMT6-bearing mice were left untreated or were *i.v.* injected with NPs@PBT on -12 h and tumors were performed with PTT (40 °C for 5 min) on -0.5 h. DiD-labelled neutrophils were incubated with DiO-labelled PEG-*b*-PLGA NPs (NPs@DiO) and intratumorally injected into the mice at 0 h. After 20 h, DiO fluorescent signals were analyzed by flow cytometry. **(a)** Flow cytometric measurements of percentage of DiD-labelled neutrophils that contained NPs@DiO. **(b)** Flow cytometric measurements and **(c)** corresponding quantitative analysis of percentage of tumor cells that internalized NPs@DiO released from transferred neutrophils. *n* = 4 per group. Data are shown as mean ± SEM and analyzed by unpaired Student's *t*-test. *****P* < 0.0001. Source data are provided as a Source Data file.

Please also refer to our response to Q#3 of Reviewer #1. We performed additional experiments using EGFP expressing EMT6 tumor cells (EMT6-EGFP) to verify that the cargos released by neutrophils could be taken up by tumor cells. In PTT-treated EMT6-EGFP tumors, more NPNs@DiD were observed within tumor cells (**Supplementary Fig. 30**), proving that tumor cells could internalize the NPNs released from neutrophils. These data was included in the supplementary information as **Supplementary Fig. 30** in the revision and relevant discussion was added in the revised manuscript (page 15, line 15-17).

Supplementary Fig. 30 EGFP expressing EMT6 tumor cells (EMT6-EGFP) internalized DiD-labelled NPNs (NPNs@DiD) released by neutrophils in the PTT-treated tumors. EMT6-EGFP tumor bearing mice were left untreated or were *i.v.* injected with NPs@PBT. After 12 h, the tumors were treated with PTT (40 °C for 5 min). After another 30 min, neutrophils containing NPNs@DiD were intratumorally injected into the mice. And 9 h later, the tumors were excised and sliced for fluorescence microscopy imaging. Scale bars, 20 μ m for the amplification panels, and 100 μ m for all other panels.

5. Considering that LPS is injected IV, levels of serum cytokines, such as IL-6, TNF- α , etc should be measured. The toxicity appears minor but this would nonetheless be important to describe.

Author reply: We thank the reviewer for his/her valuable suggestions. According to the suggestion, we measured the levels of pro-inflammatory cytokines IL-6 and TNF- α in serum of mice administrated with PBS, NPs, or NPNs. Serum levels of proinflammatory cytokines IL-6 and TNF- α were significantly increased at 4 h post injection of NPNs but dropped to normal levels at 24 h, indicating that NPNs alone only induced a transient inflammation. The data was included in the supplementary information as **Supplementary Fig. 19** in the revision and relevant discussion was added in the revised manuscript (pages 13, line 4-7).

Supplementary Fig. 19 Concentrations of pro-inflammatory cytokines IL-6 (**a**) and TNF- α (**b**) in serum at the indicated time points post injection of PBS, NPs, and NPNs. Data are shown as mean \pm SEM and analyzed by unpaired Student's *t*-test. $n = 3$ per group. * $P < 0.05$, ** $P < 0.01$, *** $P < 0.001$. *n.s.*, not significant. Source data are provided as a Source Data file.

Reviewer #3 (Remarks to the Author):

Recommendation: Publish in Nature Communications after minor revisions noted.

In this manuscript, Li et al. established a pathogen-mimicking nano-pathogenoid (NPN) system to hitchhike circulating neutrophils in vivo for tumor-targeted drug delivery. They showed that the circulating neutrophils were able to accumulate and penetrate in tumors with photothermal therapy (PTT) enhanced inflammation in response to increased level of chemokines such as CXCL1. They also demonstrated that the neutrophils in circulation could internalize NPN which were transported to the tumors and released in situ partially during NETosis of the neutrophils. When cisplatin was loaded within the NPN and used in combination with PTT, increased tumor growth inhibition was observed after a single dosage, and tumor regression was achieved after repeated treatments. This bio-inspired strategy provided a novel strategy for neutrophil targeting in vivo and exhibited great potential and effectiveness in cancer therapy, which could be also possibly used in treating infectious and autoimmune diseases. In general, the present work merit publication in Nature Communications after addressing the following minor issues.

Author reply: We thank this reviewer for his/her very positive assessment on this work and we have carefully addressed the issues raised by the reviewer.

1. Successful coating of OMVs onto nanoparticles is a prerequisite for neutrophils to capture and deliver NPs. There was no related data demonstrating the structure of NPN in the manuscript. Please provide.

Author reply: We thank the reviewer for his/her valuable suggestions. Accordingly, we used transmission electron microscope (TEM) to observe the structure of NPNs. TEM images of NPNs negatively stained with 1% phosphotungstic acid showed a clear shell surrounding the NPs, proving successful coating of OMVs onto NPs (**Supplementary Fig. 15**). This data was included in the supplementary information as **Supplementary Fig. 15** in the revision and relevant discussion was added in the revised manuscript (page 11, line 21, 22 and page 12, line 1).

Supplementary Fig. 15 A typical TEM image of NPNs, which showed the core-shell structure. Scale bar, 50 nm.

2. Extrinsic particles internalized by neutrophils usually undergo destruction and deactivation. Please verify the cytotoxicity of naïve NPNs and NPNs released by neutrophils.

Author reply: We thank the reviewer for his/her valuable suggestions. We performed experiments to evaluate the cytotoxicity of naïve NPNs and NPNs released by neutrophils. Neutrophils were incubated with NPNs@Pt and then treated with PMA to trigger the disruption of neutrophil membrane and subsequent release of NPNs@Pt. The cytotoxicity of released NPNs@Pt against

EMT6 cells was compared with naïve NPNs@Pt by a MTT assay. The results showed that NPNs@Pt released from neutrophils still maintained their cytotoxicity, which was comparable with that of NPNs@Pt. This data was included in the supplementary information as **Supplementary Fig. 27** in the revision and relevant discussion was added in the revised manuscript (page 14, line 21, 22, and page 15, line 1, 2).

Supplementary Fig. 27 Cytotoxicity of naïve NPNs@Pt and NPNs@Pt released from PMA-treated neutrophils against EMT6 cells. Neutrophils containing NPNs@Pt were treated with 100 nM PMA for 4 h. Then naïve NPNs@Pt or the supernatant containing NPNs@Pt released from neutrophils was added to EMT6 cells at the cisplatin concentration of 15 $\mu\text{g}/\text{mL}$. After 24 h, the viability of EMT6 cells was detected using the MTT assay. $n = 8$ for NPNs@Pt group and $n = 9$ for the other two groups. Data are shown as mean \pm SEM and analyzed by unpaired Student's t -test. **** $P < 0.0001$. *n.s.*, not significant. Source data are provided as a Source Data file.

3. Please verify whether the migration of neutrophils was affected by captured NPNs, as this capability is essential for tumor-targeted drug delivery.

Author reply: We thank the reviewer for his/her valuable suggestions. To verify whether NPNs uptake could affect the recruitment of neutrophils to inflamed tumors, EMT6 bearing mice were treated with PTT and then *i.v.* injected with PBS, NPs, or NPNs. After 4 h, flow cytometry analyses of the ratio of neutrophils in the tumors (*i.e.*, relative to the tumor cells) showed that NPNs uptake did not change the tumor homing of neutrophils, as evident by the similar ratio of neutrophils to tumor cells among different groups (**Supplementary Fig. 18**). This data was included in the supplementary information as **Supplementary Fig. 18** in the revision and relevant discussion was added in the revised manuscript (page 12, line 22 and page 13, line 1-4).

Supplementary Fig. 18 Uptake of NPNs by neutrophils did not alter neutrophil infiltration into tumors. EMT6 bearing mice were treated with PTT and then *i.v.* injected with PBS, PEG-*b*-PLGA NPs, or NPNs. After 4 h, the ratio of neutrophils/tumor cells in the tumors was analyzed by flow cytometry. $n = 5$ for untreated group and $n = 3$ for the other three groups. Data are shown as mean \pm SEM and analyzed by unpaired Student's *t*-test. *n.s.*, not significant. Source data are provided as a Source Data file.

4. PTT at a higher temperature such as 42 °C would exert a stronger cell eradication effect and be more efficient recruitment of neutrophils (Fig. 2j). What is the rationale of maintaining intratumoral temperature at ~40 °C during in vivo study? Please discuss.

Author reply: We thank the reviewer for his/her valuable suggestions. We agree with this reviewer in that a higher temperature would exert a stronger cell eradication effect and would recruit more neutrophils. In our studies, PTT of 40 °C was mainly used for two reasons. First, although the 42 °C PTT treatment induced more tumor-infiltration of neutrophils compared with 40 °C, the difference was not statistically significant. That is being said, a mild laser irradiation was already sufficient to create an acute intratumoral inflammation for neutrophil recruitment. Second, PTT of higher temperature has been reported to cause severe side effect to normal tissues near the lesions (*Nat. Commun.*, **7** 10437-10446 (2016)). Put together, we used 40 °C PTT treatment instead of 42 °C in this study. We have emphasized the rationales accordingly in the revised manuscript (page 7, line 21, 22 and page 8, line 1-3)

5. In Fig. 4, intravital imaging was performed to monitor the behavior of neutrophils in the tumors in real-time. Although very convincing, the results reflected the behavior of neutrophils in tumor tissue with functional perfusion. Please discuss how did the neutrophil migration and NPN diffusion affect tumor tissues without effective perfusion.

Author reply: The reviewer raised a critical concern regarding the infiltration of neutrophils in tumor microspace with low perfusion blood vessels (*e.g.*, in the core area of solid tumors). In fact, the infiltration of neutrophil in inflammatory tissues is a cascaded process involving multiple consecutive steps at least including migration in blood vessel, extravasation, and interstitial diffusion. In low perfusion tumor regions featured with collapsed and afunctional vascular structure, the migration in blood vessel and extravasation could be limited. However, neutrophils

could still actively migrated into those areas following the chemokine gradient based on previous reports (*Curr Opin Cell Biol*, **25** 650-658 (2013); *Trends Immunol*, **37** 273-286 (2016)). The relevant discussion was added in the revised manuscript (page 20 line 19-22, and page 21, line 1).

6. In Fig. 8j and supplementary Fig 19, the result showed that OMV injection resulted in a transient weight loss of mice, indicating some toxicity of OMV, how to solve the problem? Please comment.

Author reply: The transient weight loss during OMVs treatments might be caused by the acute enteritis induced by the PAMPs on OMVs, mainly lipopolysaccharide (LPS) (*Journal of Clinical Investigation*, **128** 267-280 (2018)). The toxicity issue can be addressed by using bacteria strains with lower pathogenicity or genetically modified attenuated bacteria strains, for example, with impaired LPS signaling (*Nat. Commun.*, **8** 626-634 (2017); *ACS Nano*, **12** 5995-6005 (2018)). We have discussed this in the revision (page 22, line 13-17). The relevant references were added as ref 40 and 55 in the revised manuscript.

REVIEWERS' COMMENTS:

Reviewer #1 (Remarks to the Author):

The authors have done a magnificent work in addressing all my suggestion and have provided an impressive amount of data. In my opinion the manuscript fully deserves to be published in Nature Communication with higher priority.

Reviewer #2 (Remarks to the Author):

The authors have adequately responded to all of the reviewer comments. This is a very interesting, rigorous, and well executed paper with potential to impact drug delivery to tumors.

Reviewer #3 (Remarks to the Author):

The authors have successfully addressed all the reviewer's comments, and the quality of the manuscript has been significantly improved. It can now be published in the Nature Communications.

REVIEWERS' COMMENTS:

Reviewer #1 (Remarks to the Author): The authors have done a magnificent work in addressing all my suggestion and have provided an impressive amount of data. In my opinion the manuscript fully deserves to be published in Nature Communication with higher priority.

Author reply: We thank this reviewer for his/her encouraging comments on this work.

Reviewer #2 (Remarks to the Author): The authors have adequately responded to all of the reviewer comments. This is a very interesting, rigorous, and well executed paper with potential to impact drug delivery to tumors.

Author reply: We thank this reviewer for his/her very positive comments on this work.

Reviewer #3 (Remarks to the Author): The authors have successfully addressed all the reviewer's comments, and the quality of the manuscript has been significantly improved. It can now be published in the Nature Communications.

Author reply: We thank this reviewer for his/her approval of this work.